# Learning Shape-Independent Transformation via Spherical Representations for Category-Level Object Pose Estimation

**Huan Ren**[1,2]  **Wenfei Yang**[1,2,3]  **Xiang Liu**[4]  **Shifeng Zhang**[5]  **Tianzhu Zhang**[1,2*]

[1]University of Science and Technology of China
[2]National Key Laboratory of Deep Space Exploration, Deep Space Exploration Laboratory
[3]Jianghuai Advance Technology Center   [4]Dongguan University of Technology   [5]Sangfor Technologies
rh_hr_666@mail.ustc.edu.cn   {yangwf,tzzhang}@ustc.edu.cn

## Abstract

Category-level object pose estimation aims to determine the pose and size of novel objects in specific categories. Existing correspondence-based approaches typically adopt point-based representations to establish the correspondences between primitive observed points and normalized object coordinates. However, due to the inherent shape-dependence of canonical coordinates, these methods suffer from semantic incoherence across diverse object shapes. To resolve this issue, we innovatively leverage the sphere as a shared proxy shape of objects to learn shape-independent transformation via spherical representations. Based on this insight, we introduce a novel architecture called SpherePose, which yields precise correspondence prediction through three core designs. Firstly, We endow the point-wise feature extraction with $SO(3)$-invariance, which facilitates robust mapping between camera coordinate space and object coordinate space regardless of rotation transformation. Secondly, the spherical attention mechanism is designed to propagate and integrate features among spherical anchors from a comprehensive perspective, thus mitigating the interference of noise and incomplete point cloud. Lastly, a hyperbolic correspondence loss function is designed to distinguish subtle distinctions, which can promote the precision of correspondence prediction. Experimental results on CAMERA25, REAL275 and HouseCat6D benchmarks demonstrate the superior performance of our method, verifying the effectiveness of spherical representations and architectural innovations.

## 1 Introduction

Object pose estimation, which involves predicting the 3D rotation $R \in SO(3)$ and 3D translation $t \in \mathbb{R}^3$ of observed objects, has received considerable attention from the research community due to its crucial applications in augmented reality (Marchand et al., 2015; Su et al., 2019), robotic manipulation (Liu et al., 2023a; Wen et al., 2022), and hand-object interaction (Lin et al., 2023c), etc. While many prior instance-level object pose estimation methods (Peng et al., 2019; Wang et al., 2019a; 2021a) have achieved promising performance, their dependence on CAD models restricts the generalization ability. To mitigate this problem, category-level object pose estimation has been introduced in Wang et al. (2019b), which aims to reason about the 6D pose $\{R, t\} \in SE(3)$ and 3D size $s \in \mathbb{R}^3$ of unseen objects in specific categories, without the need for their CAD models.

Existing category-level methods can be mainly divided into two groups, i.e., direct regression-based methods (Chen et al., 2021; Di et al., 2022; Lin et al., 2023b) and correspondence-based methods (Wang et al., 2019b; Tian et al., 2020; Liu et al., 2023b; Lin et al., 2024). The former approaches seek to directly regress object pose in an end-to-end manner. Although conceptually simple, they struggle with the pose-sensitive feature learning due to the non-linearity of the entire pose search space $SE(3)$ (Lin et al., 2022a; 2023b). In contrast, the latter methods focus on establishing the correspondence between camera coordinate space and the Normalized Object Coordinate Space

---

*Corresponding author. Project page: https://renhuan1999.github.io/SpherePose.

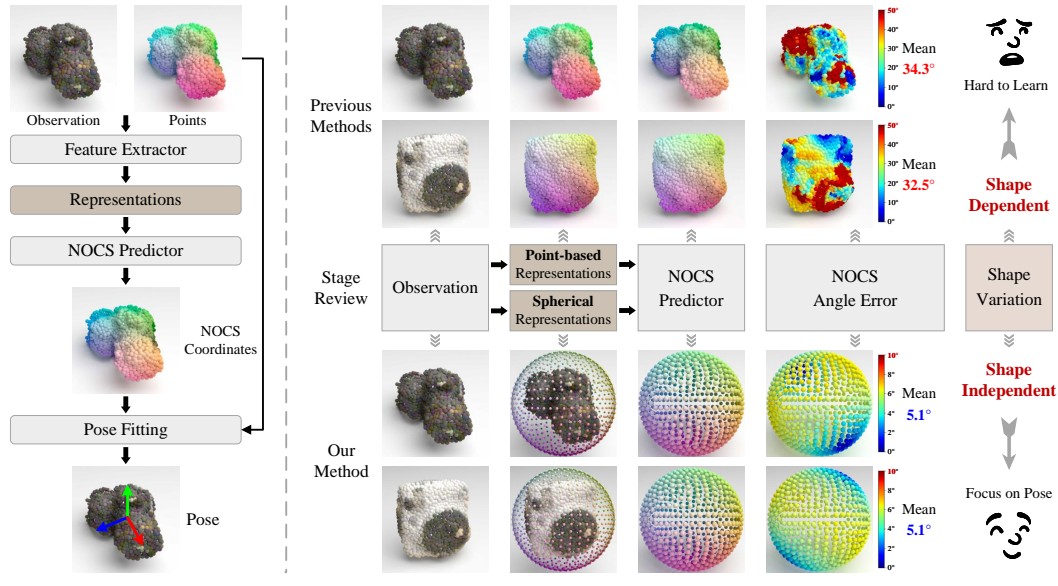

(a) Correspondence-based Pipeline      (b) Comparison Between Our Method and Previous Methods

Figure 1: We propose a novel approach for category-level object pose estimation based on spherical representations. (a) The classical pipeline of correspondence-based methods, where representations denote the organization form of observation data, e.g., point-based representations in $\mathbb{R}^3$, which is straightforward and commonly adopted in previous methods. (b) The overview of comparison between our method and previous methods. Note that each XYZ position of representations and NOCS coordinates is visualized as an RGB tuple. Our method employs **spherical representations** to learn shape-independent transformation, yielding smaller NOCS angle errors compared to DPDN (Lin et al., 2022b), which adopts **point-based representations** and suffers from the shape-dependence.

(NOCS) (Wang et al., 2019b), and then solve the pose through the Umeyama algorithm (Umeyama, 1991) or deep estimators (Lin et al., 2022b), as illustrated in Figure 1(a). We follow this paradigm, where the correspondences in the linear coordinate search space $\mathbb{R}^3$ are easier to learn.

Nevertheless, existing correspondence-based methods typically adopt *point-based representations*, where discrete points back-projected from observed depth maps serve as the representations of observation data. Subsequently, the NOCS coordinates of these points are derived by geometric normalization and alignment of object shapes, which are inherently **shape-dependent**. As a result, observed points with similar semantics across diverse object shapes are mapped to distinct NOCS coordinates, also known as semantic incoherence in Wan et al. (2023). For example, points on the camera lens are mapped to different coordinates in NOCS if the camera length is different. Since the correspondence prediction need to take into account both pose information and large shape variation, it is hard to learn and generalize, which results in large NOCS angle errors as demonstrated in Figure 1(b)*(Top)*. To resolve the above issue, we propose to innovatively leverage the sphere as a shared proxy shape of objects to learn **shape-independent** transformation via *spherical representations*. Specifically, we first uniformly divide the sphere with the Hierarchical Equal Area iso-Latitude Pixelation (HEALPix) grids (Gorski et al., 2005), each of which is represented by its center anchor. Then observed points are projected onto the spherical grids, with point-wise features assigned to the corresponding spherical anchors, which serves as the new representations of observation. Since these spherical anchors are coherent across various object shapes, the spherical NOCS coordinates are shape-independent, and thus more precise correspondence prediction can be achieved by focusing only on the pose information, as shown in Figure 1(b)*(Bottom)*.

Based on the sphere representations, we introduce a novel architecture for category-level object pose estimation, termed **SpherePose**. Given a centralized and scale-normalized observation[1], point-wise features are first extracted and then assigned to spherical anchors with HEALPix spherical projection, yielding the spherical representations. The shape-independent transformation is subsequently

---

[1]Note that since translation and size are relatively easier to be estimated, we follow VI-Net (Lin et al., 2023b) to leverage a lightweight PointNet++ for prediction, and focus on more challenging rotation estimation.

learned between the spherical anchors and the spherical NOCS coordinates for rotation estimation. In order to boost the precision of correspondence prediction, we further present three core designs. **The first design is the SO(3)-invariant point-wise feature extraction.** It is intuitive that no matter how an object is rotated, the feature of a specific point on that object should remain invariant as it is mapped to a static coordinate in NOCS. Driven by this insight, for deep image features, we employ pretrained DINOv2 (Oquab et al., 2024) features with patch-wise consensus semantics, which are robust to rotations (Chen et al., 2024). As for deep point cloud features, we propose the Color-PointNet++ network by making minimal adjustments to PointNet++ (Qi et al., 2017), which derives input features from RGB values rather than absolute XYZ coordinates, inherently ensuring SO(3)-invariance by focusing on the color information and relative proximity of point cloud. **The second design is the spherical feature interaction.** Due to self-occlusion, depth cameras fail to perceive the occluded regions on the backside of objects, resulting in incomplete point cloud. Consequently, after projecting point-wise features onto the sphere, there exist numerous spherical anchors with no assigned features. To this end, we adopt the attention mechanism (Vaswani, 2017) with learnable position embedding to facilitate the interaction and propagation of features among anchors, where initially unassigned spherical features can be reasoned out. Moreover, by holistically integrating the relationship among spherical anchors, we acquire comprehensive spherical features which are robust to noise. **The third design is a hyperbolic correspondence loss function.** For the supervision of correspondences, we derive the gradients of several typical loss functions and realize that they exhibit minor gradients around zero, which is prone to yield ambiguous prediction. To achieve more precise prediction, we leverage the computation of correspondence distance in hyperbolic space (Lin et al., 2023a), which yields higher gradients near zero and can discern more subtle distinctions.

To sum up, the main contributions of this paper are as follows:

- We present an innovative approach that utilizes the sphere as a proxy shape of objects to learn shape-independent transformation via HEALPix spherical representations, which addresses the semantic inconsistency of shape-dependent canonical coordinates that plagues existing correspondence-based methods with point-based representations.

- We introduce an architecture for category-level object pose estimation that achieves precise correspondence prediction through three core designs: SO(3)-invariant point-wise feature extraction, spherical feature interaction, and a hyperbolic correspondence loss function.

- Extensive experiments on existing benchmarks demonstrate the superior performance of our method over state-of-the-art approaches, verifying the effectiveness of spherical representations and architectural innovations.

## 2 RELATED WORK

The task of category-level object pose estimation is first introduced in Wang et al. (2019b) to predict the 6D pose and size of unseen objects in a given category, and existing methods can be categorized into two groups, i.e., direct regression-based methods and correspondence-based methods.

**Direct Regression-based Methods.** This category of approaches intends to directly regress the object pose in an end-to-end manner. FS-Net (Chen et al., 2021) proposes to decouple the rotation into two perpendicular vectors that simplifies the prediction, and utilizes a 3D graph convolution (3D-GC) autoencoder for feature extraction. GPV-Pose (Di et al., 2022) enhances the learning of pose-sensitive features with geometric consistency, and HS-Pose (Zheng et al., 2023) further extends 3D-GC to extract hybrid scope latent features that combine both global and local geometric information. VI-Net (Lin et al., 2023b) presents a novel rotation estimation network that decouples the rotation into a viewpoint rotation and an in-plane rotation, simplifying the task by leveraging spherical representations. Based on the decoupled rotation, SecondPose (Chen et al., 2024) extracts SE(3)-consistent semantic and geometric features to further enhance the pose estimation. However, these methods struggle with the pose-sensitive feature learning due to the non-linearity of SE(3).

**Correspondence-based Methods.** This group of approaches aims to establish the correspondence between camera coordinate space and object coordinate space, and then acquire the pose through pose fitting algorithm, such as Umeyama (Umeyama, 1991). As a cornerstone, the Normalized Object Coordinate Space (NOCS) is introduced in Wang et al. (2019b), which serves as a shared canonical representation to align object instances within a given category. Later, SPD (Tian et al., 2020) proposes to handle the intra-class shape variation by reconstructing the 3D object model from

a pre-learned categorical shape prior, and then match the observed point cloud to the reconstructed 3D model. Inspired by it, CR-Net (Wang et al., 2021b) and SGPA (Chen & Dou, 2021) are presented to improve the reconstruction quality. DPDN (Lin et al., 2022b) learns the shape prior deformation in the feature space, and replaces the traditional Umeyama algorithm with a deep estimator to directly regress the pose and size. Query6DoF (Wang et al., 2023) and IST-Net (Liu et al., 2023b) eliminate the dependence on explicit shape priors with implicit counterparts and implicit space transformation, respectively. More recently, AG-Pose (Lin et al., 2024) proposes to detect a set of sparse keypoints to represent the geometric structures of objects and establish robust keypoint-level correspondences for pose estimation. Although these methods have demonstrated strong performance, they all adopt point-based representations, where the NOCS coordinates are derived by geometric normalization and alignment of object shapes. When handling diverse object shapes, the NOCS coordinates become semantically incoherent, which are hard to learn and generalize. To tackle this issue, Semantically-aware Object Coordinate Space (SOCS) is presented in Wan et al. (2023), where semantically meaningful SOCS coordinates are built by keypoint-guided non-rigid object alignment to the categorical average shape. However, due to the non-rigid alignment process, there is a mismatch between the geometric structure of the observed point cloud and the SOCS coordinates. This discrepancy plagues the pose fitting, as the transformation may not be optimal for all parts of the object. In this work, we instead resort to spherical representations, where the sphere is leveraged as a shared proxy shape of objects to learn shape-independent transformation between spherical anchors and spherical NOCS coordinates. Thus precise correspondence prediction can be achieved without the need to account for large shape variation, thereby enhancing subsequent pose estimation.

## 3 METHOD

**Target.** Given an RGB-D image, we first acquire the instance segmentation masks with an offline Mask R-CNN (He et al., 2017), which then yield the cropped RGB image $I \in \mathbb{R}^{H \times W \times 3}$ and segmented depth image for each instance. Partially observed point cloud $P^O \in \mathbb{R}^{N \times 3}$ is derived by back-projecting and downsampling the segmented depth image, where $N$ denotes the number of points. With $I$ and $P^O$ as inputs, our method predicts the rotation $R \in \mathrm{SO}(3)$, translation $t \in \mathbb{R}^3$, and size $s \in \mathbb{R}^3$ of the observed instance.

**Overview.** As illustrated in Figure 2, the proposed method mainly consists of two components. First, in the spherical representation projection phase (Section 3.1), point-wise features are extracted and then assigned to spherical anchors with HEALPix spherical projection, yielding the spherical representations. Subsequently, in the spherical rotation estimation phase (Section 3.2), spherical anchors perform feature interaction with each other through the attention mechanism, and then they are mapped to the spherical NOCS coordinates for rotation estimation. In Section 3.3, we elaborate on the training and testing workflow, including additional translation and size prediction for complete category-level object pose estimation. Finally, in Section 3.4, we emphasize the distinctions between our method and some relevant works on spherical representations and the proxy shape.

### 3.1 SPHERICAL REPRESENTATION PROJECTION

**SO(3)-invariant Point-wise Feature Extraction.** Taking image $I$ and normalized points $P^2$ as inputs, we start with point-wise feature extraction. Note that for point-wise image features, we first employ a 2D feature extractor to $I$ and then leverage $P$ to perform point-wise selection on the extracted features. To facilitate the transformation between camera coordinate space and object coordinate space, we recognize that the point-wise features $F^P \in \mathbb{R}^{N \times C}$ should be SO(3)-invariant:

$$\psi_G(F^P) = F^P, \quad \forall G \in \mathrm{SO}(3), \tag{1}$$

where $\psi_G$ indicates the transformation of features when rotating points $P$ by rotation $G$. Intuitively, the feature of a particular point on an object should remain invariant regardless of the object's rotation, given that the point is mapped to a fixed coordinate in NOCS. To this end, we synthesize the features of observation with SO(3)-invariance from a comprehensive perspective. Specifically, for image features, we employ RGB values with low-level texture information and the pretrained DINOv2 (Oquab et al., 2024) features with high-level semantic information. Owing to the pretraining on large-scale datasets, DINOv2 possesses robust generalization capabilities and can provide

---

[2] $P \in \mathbb{R}^{N \times 3}$ is centralized and scale-normalized from $P^O$ with a lightweight PointNet++ (Qi et al., 2017).

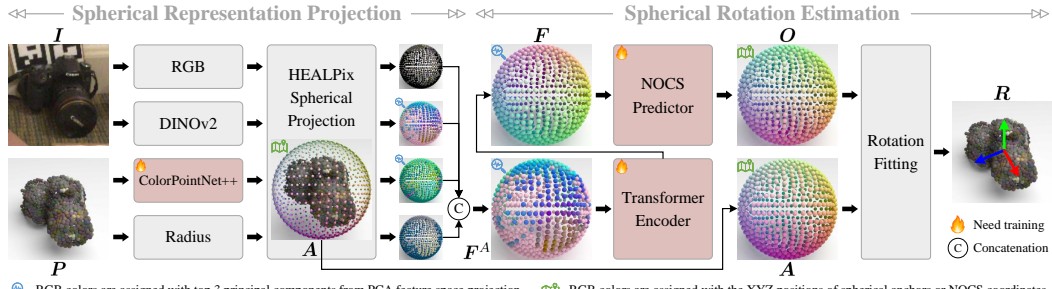

Figure 2: Overview of the proposed SpherePose. Given the observation $\boldsymbol{I}$ and $\boldsymbol{P}$, we first extract SO(3)-invariant point-wise features from four distinct perspectives and assign them to the spherical anchors $\boldsymbol{A}$ with HEALPix spherical projection, yielding initial spherical features $\boldsymbol{F}^A$. Then, the Transformer encoder module is employed for spherical feature interaction and integrates comprehensive spherical features $\boldsymbol{F}$. Finally, we predict the corresponding spherical NOCS coordinates $\boldsymbol{O}$ via a NOCS predictor, which is applied to the estimation of rotation $\boldsymbol{R}$.

semantically consistent patch-wise features that are robust to rotations (Chen et al., 2024). As for point cloud features, we adopt radius[3] values with low-level spatial information, along with features with high-level geometric information from the proposed ColorPointNet++, which is a variant of PointNet++ (Qi et al., 2017). The input features of the original PointNet++ are derived from absolute XYZ coordinates, which are inherently SO(3)-equivariant. To validate the importance of SO(3)-invariance for correspondence prediction, we make minimal adjustments to adapt the architecture for extracting SO(3)-invariant features, by replacing raw XYZ coordinates with RGB values. Given the SO(3)-invariance of color attributes and kNN operation in grouping and interpolating, our ColorPointNet++ yields SO(3)-invariant point-wise features, as detailed in Appendix A.1. The final point-wise features $\boldsymbol{F}^P \in \mathbb{R}^{N \times C}$ are the concatenation of all four point-wise features that capture distinct information from the observation, followed by feature dimension reduction to $C$.

**HEALPix Spherical Projection.** Once point-wise features are extracted, previous methods adopt point-based representations to directly predict the NOCS coordinates corresponding to the observed points, which causes semantic incoherence across diverse object shapes, as introduced in Section 1. Instead, we propose to employ the sphere as a shared proxy shape of objects to learn shape-independent transformation via spherical representations. Specifically, we first uniformly divide the sphere with HEALPix grids (Gorski et al., 2005) as illustrated in Figure 4(b), each of which is equal in area and represented by its center anchor $\boldsymbol{A}_m \in \mathbb{R}^3, m \in \{1, \ldots, M\}$, where $M$ denotes the number of grids. We then project the normalized points $\boldsymbol{P}$ onto the spherical grids, and assign point-wise features $\boldsymbol{F}^P$ to the corresponding spherical anchors, which serve as the new representations of observation with spherical features $\boldsymbol{F}^A \in \mathbb{R}^{M \times C}$. In detail, consider a grid indexed by $m$, we follow VI-Net (Lin et al., 2023b) to search within this grid region for the point with the largest radius value, denoted as $\boldsymbol{P}_n$, and assign its feature $\boldsymbol{F}_n^P$ to the anchor $\boldsymbol{A}_m$. If there is no point projected onto the grid, we set $\boldsymbol{F}_m^A = \boldsymbol{0}$. Since these spherical anchors are coherent across diverse objects, the corresponding spherical NOCS coordinates are shape-independent.

### 3.2 Spherical Rotation Estimation

**Spherical Feature Interaction.** Due to self-occlusion, depth cameras fail to perceive the occluded regions on the backside of objects, causing incomplete point cloud. This phenomenon deteriorates when projected onto spherical grids, leaving almost half of the spherical anchors without assigned features. To this end, we employ a Transformer encoder with attention mechanism (Vaswani, 2017) to facilitate feature interaction and propagation among spherical anchors $\boldsymbol{A} \in \mathbb{R}^{M \times 3}$ in a global perspective, yielding comprehensive spherical features $\boldsymbol{F} \in \mathbb{R}^{M \times C}$ from the initial spherical features $\boldsymbol{F}^A$ with a sequence of self-attention layers and MLP blocks, formulated as follows:

$$\begin{aligned} \boldsymbol{F}^{l+1} &= \mathrm{MLP}(\hat{\boldsymbol{F}}^{l+1}) + \hat{\boldsymbol{F}}^{l+1}, \\ \hat{\boldsymbol{F}}^{l+1} &= \mathrm{Attention}(\boldsymbol{F}^l) + \boldsymbol{F}^l, \end{aligned} \qquad l \in \{0, \ldots, L-1\}, \qquad (2)$$

---

[3]Radius refers to the Euclidean distance from the origin, i.e., the $L^2$ norm of 3D point's coordinates.

where $L$ is the number of layers and $\boldsymbol{F} = \boldsymbol{F}^L$, $\boldsymbol{F}^0 = \boldsymbol{F}^A$. Due to the disorder of spherical anchors, the learnable position embeddings $\boldsymbol{E}^{pos} \in \mathbb{R}^{M \times C}$ are assigned to corresponding anchors and contribute to the computation of attention for feature propagation. In formula, the self-attention in $l$-th layer is defined as:

$$\text{Attention}(\boldsymbol{F}^l) = \text{Softmax}\left(\frac{(\boldsymbol{Q}^l \boldsymbol{W}^Q)(\boldsymbol{K}^l \boldsymbol{W}^K)^\top}{\sqrt{C}}\right)(\boldsymbol{V}^l \boldsymbol{W}^V), \tag{3}$$

$$\boldsymbol{Q}^l = \boldsymbol{F}^l + \boldsymbol{E}^{pos}, \quad \boldsymbol{K}^l = \boldsymbol{F}^l + \boldsymbol{E}^{pos}, \quad \boldsymbol{V}^l = \boldsymbol{F}^l, \tag{4}$$

where $W^{Q/K/V} \in \mathbb{R}^{C \times C}$ denotes learnable projection matrix for query, key and value, respectively, with independent parameters across layers. Through global feature interaction and propagation, those spherical features that are initially unassigned can be reasoned out. Moreover, by attending to the relationship among spherical anchors from a holistic perspective, we integrate pose-coherent spherical features which can alleviate the interference of noise.

**Correspondence-based Rotation Estimation.** Given the spherical features $\boldsymbol{F}$, we follow previous methods (Lin et al., 2022b; 2024) to first predict the spherical NOCS coordinates $\boldsymbol{O} \in \mathbb{R}^{M \times 3}$ corresponding to spherical anchors $\boldsymbol{A}$ with an MLP-based NOCS predictor. Since the spherical anchors and NOCS coordinates are on the unit sphere, we $L^2$ normalize the output of MLP:

$$\boldsymbol{O}_m = \frac{\text{MLP}(\boldsymbol{F}_m)}{\|\text{MLP}(\boldsymbol{F}_m)\|_2}, \quad m \in \{1, \dots, M\}. \tag{5}$$

After acquiring the correspondences between spherical anchors and spherical NOCS coordinates, there are several approaches to acquire the rotation $\boldsymbol{R} \in \text{SO}(3)$, such as the Umeyama algorithm (Umeyama, 1991) or deep estimators (Lin et al., 2022b). We adopt the Umeyama algorithm with RANSAC (Fischler & Bolles, 1981) for outlier removal as it does not require extra training.

### 3.3 CATEGORY-LEVEL OBJECT POSE ESTIMATION

**Training.** For the estimation of translation $\boldsymbol{t} \in \mathbb{R}^3$ and size $\boldsymbol{s} \in \mathbb{R}^3$, we follow VI-Net (Lin et al., 2023b) to employ a simple and lightweight PointNet++ (Qi et al., 2017). It takes the observed point cloud $\boldsymbol{P}^O$ and the point-wise selected RGB attributes from image $\boldsymbol{I}$ as input, and directly regresses $\boldsymbol{t}$ and $\boldsymbol{s}$ with $L^1$ loss as follows:

$$\mathcal{L}_{\boldsymbol{ts}} = \|\boldsymbol{t} - \boldsymbol{t}^{gt}\|_1 + \|\boldsymbol{s} - \boldsymbol{s}^{gt}\|_1. \tag{6}$$

As for the estimation of rotation $\boldsymbol{R} \in \text{SO}(3)$, we adopt the correspondence-based paradigm to supervise the spherical NOCS coordinates $\boldsymbol{O}$. Specifically, we generate ground truth spherical NOCS coordinates $\boldsymbol{O}^{gt}$ by mapping spherical anchors $\boldsymbol{A}$ into NOCS using the ground truth rotation $\boldsymbol{R}^{gt}$:

$$\boldsymbol{O}_m^{gt} = (\boldsymbol{R}^{gt})^\top \boldsymbol{A}_m, \quad m \in \{1, \dots, M\}, \tag{7}$$

and the prediction errors $\boldsymbol{e}$ can then be measured by the distance between them, defined as:

$$\boldsymbol{e}_m = \begin{cases} \|\boldsymbol{O}_m - \boldsymbol{O}_m^{gt}\|_1, & \text{as } L^1 \text{ distance} \\ \|\boldsymbol{O}_m - \boldsymbol{O}_m^{gt}\|_2, & \text{as } L^2 \text{ distance} \end{cases}, \quad m \in \{1, \dots, M\}. \tag{8}$$

Previous methods employ either $\boldsymbol{e}$ directly as the loss function or the smooth $L^1$ loss (Wang et al., 2019b), which uses a squared term $e^2$ if $e$ fall below a certain threshold and the term $e$ otherwise. However, we examine the gradients of these loss functions in Figure 3 and discover that they exhibit minor gradients around zero, which is prone to generate ambiguous prediction. To promote the precision of correspondence prediction, we leverage the computation of distance in hyperbolic space (Lin et al., 2023a), which yields higher gradients near zero and can discern more subtle distinctions. In detail, the $\text{arcosh}(1 + x)$ function is employed to the errors $\boldsymbol{e}$ with $L^2$ distance, and the final hyperbolic correspondence loss function for $\boldsymbol{O}$ is formulated as follows:

$$\mathcal{L}_{corr} = \frac{1}{M} \sum_{m=1}^{M} \text{arcosh}(1 + \boldsymbol{e}_m) = \frac{1}{M} \sum_{m=1}^{M} \text{arcosh}\left(1 + \|\boldsymbol{O}_m - \boldsymbol{O}_m^{gt}\|_2\right). \tag{9}$$

**Inference.** Given the cropped RGB image $\boldsymbol{I}$ and point cloud $\boldsymbol{P}^O$, we first predict the translation $\boldsymbol{t}$ and size $\boldsymbol{s}$ via PointNet++, and then normalize[4] the point cloud as $\boldsymbol{P} = (\boldsymbol{P}^O - \boldsymbol{t})/\|\boldsymbol{s}\|_2$, which is fed into our SpherePose for the estimation of rotation $\boldsymbol{R}$.

---

[4]Note that the predicted $\boldsymbol{t}$ and $\boldsymbol{s}$ are used for centralization and scale-normalization in the inference stage, while the ground truth $\boldsymbol{t}^{gt}$ and $\boldsymbol{s}^{gt}$ are used instead during the training stage.

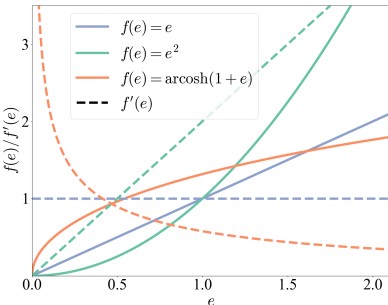

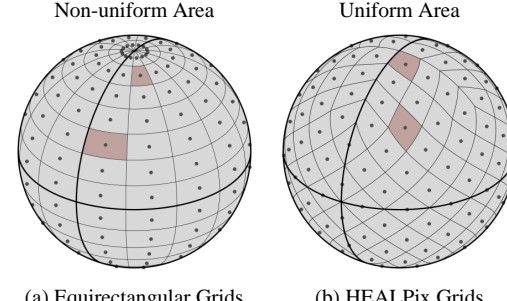

| Non-uniform Area | Uniform Area |
| --- | --- |
| (a) Equirectangular Grids | (b) HEALPix Grids |

Figure 3: Illustration of several correspondence loss functions and their gradients.

Figure 4: Overview of (a) equirectangular grids and (b) HEALPix grids.

## 3.4 DISCUSSION

**Spherical Representations.** Although several previous methods (Lin et al., 2021; 2023b; Chen et al., 2024) have also adopted spherical representations for category-level object pose estimation, there are two fundamental distinctions. Firstly, they employ the equirectangular grids (Driscoll & Healy, 1994), which are evenly divided along latitude and longitude, as shown in Figure 4(a). However, this partition of sphere is non-uniform in area, with a higher sampling density near the poles, which is suboptimal for representing the point cloud data. This characteristic is particularly detrimental for objects with diverse poses, as non-uniform sampling of distinct semantic components results in a biased distribution of sampled data. Instead, we resort to the HEALPix grids (Gorski et al., 2005), which consistently yield the uniformly sampled spherical representations across various object poses, as illustrated in Figure 4(b). This uniform spherical partition is a more appropriate choice for the capture of point cloud structures, thereby enhancing the performance of pose estimation. Secondly, they intend to directly regress the pose in an end-to-end manner, which struggle with the pose-sensitive feature learning due to the non-linearity of $SE(3)/SO(3)$. DualPoseNet (Lin et al., 2021) employs spherical convolution in the spectral domain to learn $SO(3)$-equivariant features. However, the Fourier transforms require tensors in the Fourier domain of $SO(3)$ that scale with the cube of the bandwidth, limiting the resolution. VI-Net (Lin et al., 2023b) utilizes spherical convolution in the spatial domain to learn viewpoint-equivariant features. Although it simplifies the regression process by decoupling the rotation into viewpoint rotation and in-plane rotation, the spatial convolution has a limited receptive field. Different from them, we follow with the correspondence-based paradigm to establish the correspondences between spherical anchors and spherical NOCS coordinates, which is easier to learn in the linear space $\mathbb{R}^3$. As for the spherical feature extraction, we adopt the attention mechanism to facilitate the feature interaction and propagation among spherical anchors from a holistic perspective, yielding the comprehensive spherical features.

**Proxy Shape.** SAR-Net (Lin et al., 2022a) aims to deform the category-level proxy template point cloud to align with the observed point cloud for implicitly representing its rotational state. The object rotation is then solved from the categorical and deformed template point clouds by Umeyama algorithm. Although it learns shape-independent transformation by leveraging the category-shared template shape, the representations of the proxy shape are not well exploited to facilitate correspondence prediction. Specifically, the rotation information for guidance during the deformation process is derived from a single global feature vector of the observed point cloud, which leads to difficulties with rotation-sensitive feature learning. In contrast, we resort to the sphere as a shared proxy shape to learn shape-independent transformation via spherical representations. By projecting the observed point cloud onto the spherical grids, we preserve the intact information from the observation. The comprehensive spherical features are further extracted through the interaction and integration of features among spherical anchors from a holistic perspective, which can mitigate the interference of noise and promote the correspondence prediction. The proxy shape is also incorporated in MFOS (Lee et al., 2024) for image-aligned pose representation, which transforms the pose into a 2D image by rendering the 3D coordinates of a proxy shape (e.g., a cuboid or an ellipsoid) positioned according to the pose. Since MFOS operates with only 2D images as input, the proxy shape provides a mechanism for achieving 3D geometric perception, and the object pose is recovered by solving a PnP problem from the dense 2D-3D mapping. We differ by directly learning the 3D-3D transformation of the proxy sphere between the camera and object coordinate spaces.

Table 1: Performance comparison with state-of-the-art methods on REAL275 dataset.

| | Method | Representation | $IoU_{50}$ | $IoU_{75}$ | $5°2cm$ | $5°5cm$ | $10°2cm$ | $10°5cm$ |
|---|---|---|---|---|---|---|---|---|
| Direct Regression | GPV-Pose (Di et al., 2022) | Point-based | - | 64.4 | 32.0 | 42.9 | - | 73.3 |
| | HS-Pose (Zheng et al., 2023) | Point-based | 82.1 | 74.7 | 46.5 | 55.2 | 68.6 | 82.7 |
| | GenPose (Zhang et al., 2023) | Point-based | - | - | 52.1 | 60.9 | 72.4 | 84.0 |
| | DualPoseNet (Lin et al., 2021) | Spherical | 79.8 | 62.2 | 29.3 | 35.9 | 50.0 | 66.8 |
| | VI-Net (Lin et al., 2023b) | Spherical | - | - | 50.0 | 57.6 | 70.8 | 82.1 |
| | SecondPose (Chen et al., 2024) | Spherical | - | - | 56.2 | 63.6 | 74.7 | 86.0 |
| Correspondence | NOCS (Wang et al., 2019b) | Point-based | 78.0 | 30.1 | 7.2 | 10.0 | 13.8 | 25.2 |
| | SPD (Tian et al., 2020) | Point-based | 77.3 | 53.2 | 19.3 | 21.4 | 43.2 | 54.1 |
| | SGPA (Chen & Dou, 2021) | Point-based | 80.1 | 61.9 | 35.9 | 39.6 | 61.3 | 70.7 |
| | DPDN (Lin et al., 2022b) | Point-based | 83.4 | 76.0 | 46.0 | 50.7 | 70.4 | 78.4 |
| | IST-Net (Liu et al., 2023b) | Point-based | 82.5 | 76.6 | 47.5 | 53.4 | 72.1 | 80.5 |
| | Query6DoF (Wang et al., 2023) | Point-based | 82.5 | 76.1 | 49.0 | 58.9 | 68.7 | 83.0 |
| | AG-Pose (Lin et al., 2024) | Point-based | 83.7 | **79.5** | 54.7 | 61.7 | 74.7 | 83.1 |
| | **SpherePose** | Spherical | **84.0** | 79.0 | **58.2** | **67.4** | **76.2** | **88.2** |

## 4 EXPERIMENTS

### 4.1 EXPERIMENTAL SETUP

**Datasets.** We evaluate our methods on three benchmarks including CAMERA25, REAL275 (Wang et al., 2019b) and HouseCat6D (Jung et al., 2024). CAMERA25 is a synthetic dataset that contains 275K training images and 25K testing images from 6 object categories, which are generated by rendering foreground objects with real-world backgrounds using the mixed-reality technique. REAL275 is a real-world dataset that consists of 4.3K training images of 7 scenes and 2.75K testing images of 6 scenes, which shares the same categories with CAMERA25. HouseCat6D is an emerging real-world dataset that comprises 20K training frames of 34 scenes, 3K testing frames of 5 scenes and 1.4K validation frames of 2 scenes from 10 household categories. This collection encompasses a diverse range of photometrically challenging objects, including glass and cutlery, which are captured in a comprehensive manner across various viewpoints and occlusions.

**Evaluation Metrics.** Following previous works (Wang et al., 2019b; Chen et al., 2024), we report the mean Average Precision (mAP) of $n°m$ cm for 6D pose estimation, which denotes the percentage of prediction with rotation error less than $n°$ and translation error less than $m$ cm. We also report the mAP of Intersection over Union ($IoU_x$) for 3D bounding boxes with thresholds of $x\%$.

**Implementation Details.** For a fair comparison, we employ the same instance segmentation masks as DPDN (Lin et al., 2022b) from Mask R-CNN (He et al., 2017). The number of sampled points is $N = 2,048$. For DINOv2 features, images are first cropped and resized to $224 \times 224$ and then fed into the frozen DINOv2, followed by bilinear interpolation upsampling to the original resolution for point-wise selection. After extracting four point-wise features, we concatenate them at the feature dimension, followed by a linear layer activated by GeLU (Hendrycks & Gimpel, 2016) that reduces the dimension to $C = 128$. The resolution of HEALPix grids is expressed by the parameter $N_{side}$, and we set it to 8, which results in $M = 768$ grids. The number of the Transformer encoder layers is set to $L = 6$ by default. All experiments are conducted on a single RTX3090Ti GPU with a batch size of 64 for 200K iterations. More details can be found in Appendix A.2.

### 4.2 COMPARISON WITH STATE-OF-THE-ART METHODS

**Results on REAL275 and CAMERA25 Datasets.** Table 1 shows the comparison of our method with existing direct regression-based methods and correspondence-based methods on REAL275 dataset. It should be noted that our approach represents a pioneering application of spherical representations within the correspondence-based paradigm. From the results we can see that SpherePose outperforms the prior state-of-the-art methods on the precision of 6D pose estimation. Specifically, when compared with direct regression-based methods, SpherePose surpasses SecondPose (Chen et al., 2024) by 2.0% on $5°2cm$ and 2.2% on $10°5cm$, which also employs spherical representations and DINOv2 (Oquab et al., 2024) backbone. And when compared with correspondence-based methods, SpherePose outperforms AG-Pose (Lin et al., 2024) by 3.5% on $5°2cm$ and 5.1% on $10°5cm$, which encounters the intra-class semantic incoherence arising from point-based representations. As

Table 2: Performance comparison with state-of-the-art methods on HouseCat6D dataset.

| | Method | Representation | $IoU_{25}$ | $IoU_{50}$ | $5°2cm$ | $5°5cm$ | $10°2cm$ | $10°5cm$ |
|---|---|---|---|---|---|---|---|---|
| Direct Regression | FS-Net (Chen et al., 2021) | Point-based | 74.9 | 48.0 | 3.3 | 4.2 | 17.1 | 21.6 |
| | GPV-Pose (Di et al., 2022) | Point-based | 74.9 | 50.7 | 3.5 | 4.6 | 17.8 | 22.7 |
| | VI-Net (Lin et al., 2023b) | Spherical | 80.7 | 56.4 | 8.4 | 10.3 | 20.5 | 29.1 |
| | SecondPose (Chen et al., 2024) | Spherical | 83.7 | 66.1 | 11.0 | 13.4 | 25.3 | 35.7 |
| Correspondence | AG-Pose (Lin et al., 2024) | Point-based | 81.8 | 62.5 | 11.5 | 12.0 | 32.7 | 35.8 |
| | **SpherePose** | Spherical | **88.8** | **72.2** | **19.3** | **25.9** | **40.9** | **55.3** |

for the precision of 3D bounding boxes, we achieve comparable performance. Similar results on CAMERA25 dataset can be found in Table 8, please refer to Appendix A.3 for the details.

**Results on HouseCat6D Dataset.** Table 2 provides the quantitative results of existing methods on HouseCat6D dataset. As can be seen from the table, SpherePose achieves the best performance against state-of-the-art approaches by a large margin under all metrics. Specifically, SpherePose surpasses SecondPose (Chen et al., 2024) by 6.1% on $IoU_{50}$ and 8.3% on $5°2cm$, and outperforms AG-Pose (Lin et al., 2024) by 10.3% on $IoU_{50}$ and 7.8% on $5°2cm$. When compared under the $10°5cm$ metric, our method surpasses them by almost 20% (19.6% and 19.5%, respectively). It is notable that this dataset includes objects with photometric complexities and extensive viewpoint distributions, which is extremely challenging. The significant improvements on this benchmark further demonstrate the effectiveness of the proposed method.

**Results of Correspondence Errors.** To validate the effectiveness of the proposed spherical representations compared to point-based representations in correspondence prediction, we calculate the mean NOCS errors on the REAL275 dataset. As shown in Table 3, the NOCS errors of Sphere-Pose are lower than DPDN (Lin et al., 2022b) and AG-Pose (Lin et al., 2024) on both the angle and Euclidean distance, which indicates that the proposed method achieves more accurate correspondence prediction by learning shape-independent transformation on the proxy sphere.

Table 3: Comparison of correspondence errors.

| Method | Mean NOCS Error | |
|---|---|---|
| | Angle (°) | Distance (cm) |
| DPDN (Lin et al., 2022b) | 16.62 | 10.50 |
| AG-Pose (Lin et al., 2024) | 10.36 | 5.88 |
| **SpherePose** | **4.52** | **3.89** |

### 4.3 ABLATION STUDIES

To shed more light on the superiority of our method, we conduct comprehensive ablation studies on REAL275 dataset, as detailed below.

**Efficacy of SO(3)-invariant Point-wise Feature Extraction.** Table 4 illustrates the impact of different deep point cloud backbones on the performance of object pose estimation. The original PointNet++ (Qi et al., 2017)

Table 4: Ablation studies on the deep point cloud backbone.

| Point Cloud Backbone | $5°2cm$ | $5°5cm$ | $10°2cm$ | $10°5cm$ | Parameters |
|---|---|---|---|---|---|
| None | 55.9 | 64.6 | 74.3 | 86.7 | 0 |
| PointNet++ | 56.3 | 64.7 | 73.2 | 84.4 | 1,391,104 |
| ColorPointNet++ | **58.2** | **67.4** | **76.2** | **88.2** | 1,389,664 |

inherently lacks SO(3)-invariance due to the injection of SO(3)-equivariant absolute XYZ coordinates. To validate the importance of SO(3)-invariant features for correspondence-based methods, we introduce minimal adaptations to PointNet++ to derive our ColorPointNet++, which enables the extraction of inherently SO(3)-invariant point cloud features while maintaining the architecture and the number of parameters almost unchanged. As shown in the table, with comparable number of parameters, our ColorPointNet++ outperforms the original PointNet++ by 1.9% on $5°2cm$ and 3.8% on $10°5cm$. Notably, PointNet++ performs even worse than no point cloud backbone at all, showing a 2.3% drop on $10°5cm$. These results demonstrate the effectiveness of SO(3)-invariant point-wise features for correspondence-based object pose estimation methods.

**Efficacy of Spherical Feature Interaction.** Table 5 illustrates the impact of varying numbers $L$ of the Transformer encoder layers. When the Transformer encoder is not employed, there is a significant drop in performance to 20.7% on $5°2cm$, while when two layers of encoder are applied, the performance on $5°2cm$ goes up to 53.6%. This result demonstrates the effectiveness of spherical

Table 5: Impact of the number of encoder layers.

| Encoder Layers | $5°2cm$ | $5°5cm$ | $10°2cm$ | $10°5cm$ |
|---|---|---|---|---|
| 0 | 20.7 | 23.9 | 56.2 | 65.8 |
| 2 | 53.6 | 61.3 | 74.0 | 85.8 |
| 4 | 55.7 | 64.5 | 75.2 | 86.5 |
| 6 | **58.2** | **67.4** | **76.2** | **88.2** |
| 8 | 57.9 | 66.7 | 74.9 | 86.2 |

Table 6: Impact of distinct loss functions.

| Loss Function | $5°2cm$ | $5°5cm$ | $10°2cm$ | $10°5cm$ |
|---|---|---|---|---|
| $L^1$ | 56.0 | 64.6 | 74.4 | 86.1 |
| Smooth $L^1$ | 54.6 | 63.2 | 72.6 | 83.3 |
| Hyperbolic $L^1$ | 58.0 | 66.5 | 73.5 | 85.0 |
| $L^2$ | 54.2 | 62.6 | 72.2 | 83.2 |
| Hyperbolic $L^2$ | **58.2** | **67.4** | **76.2** | **88.2** |

feature interaction, which yields comprehensive spherical features from a holistic perspective. As the number of encoder layers increases, the precision of pose estimation is also improved, reaching a saturation at six layers. Further increasing the number of encoder layers does not lead to enhanced performance, but rather increases the inference overhead. Therefore, we set $L = 6$ in our method.

**Efficacy of the Hyperbolic Correspondence Loss Function.** Table 6 presents the impact of distinct loss functions on correspondence. As for the error with $L^1$ distance in Equation 8, we conduct evaluations using $L^1$ loss, smooth $L^1$ loss and hyperbolic $L^1$ loss, respectively. Since the smooth $L^1$ loss has minor gradients near zero, it exhibits the worst performance of 54.6% on $5°2cm$. Conversely, the hyperbolic $L^1$ loss augments the gradients around zero to distinguish subtle distinctions, which improves the $5°2cm$ metric from 56.0% of the $L^1$ loss to 58.0%. With regard to the error with $L^2$ distance in Equation 8, we also derive the hyperbolic $L^2$ loss, which results in an improvement on $5°2cm$ from 54.2% of the $L^2$ loss to 58.2%. We assume that the $L^2$ distance in Euclidean space is a better measure of the correspondence errors, so we adopt the hyperbolic $L^2$ loss for correspondence prediction. For detailed formulas of these loss functions, please refer to Appendix A.2.

**Choice of Spherical Grids.** Table 7 provides the performance comparison of adopting different spherical grids. To ensure a fair comparison, we use $M = 768$ grids for both HEALPix (Gorski et al., 2005) and Super-Fibonacci Spirals (Alexa, 2022) grids. For Equirectangular (Driscoll & Healy, 1994) grids, the latitude and longitude resolution is set to $28 \times 28$, resulting in 784 grids, which is close to 768 in number. Since the HEALPix grids are area-uniform for the partition of sphere and can better capture the structure of point cloud, it achieves 2.5% and 4.6% higher performance on the $5°2cm$ and $10°5cm$ metrics, respectively, compared to the Equirectangular grids. In addition, we experiment with other nearly uniform spherical partitions, such as Super-Fibonacci Spirals, which also outperforms Equirectangular grids with a 3.6% gain on $10°5cm$, though still slightly below the performance of HEALPix.

Table 7: Impact of the choice of spherical grids.

| Spherical Grids | $5°2cm$ | $5°5cm$ | $10°2cm$ | $10°5cm$ |
|---|---|---|---|---|
| HEALPix | **58.2** | **67.4** | **76.2** | **88.2** |
| Equirectangular | 55.7 | 64.7 | 72.1 | 83.6 |
| Super-Fibonacci Spirals | 56.5 | 65.0 | 75.8 | 87.2 |

## 5 CONCLUSION

In this work, we investigate the prevalent issue of intra-class semantic inconsistency of the canonical coordinates within existing correspondence-based approaches for category-level object pose estimation. We attribute this issue to shape-dependent point-based representations and propose to leverage the sphere as a shared proxy shape of objects to learn shape-independent transformation via spherical representations. In light of this insight, we introduce SpherePose, a novel architecture that achieves precise correspondence prediction through three core designs: SO(3)-invariant point-wise feature extraction for robust mapping between camera and object coordinate space, spherical feature interaction for holistic relationship integration, and a hyperbolic correspondence loss function for diagnosis of subtle correspondence errors. Experimental evaluations on existing benchmarks demonstrate the superiority of our method over state-of-the-art approaches.

**Limitations and Future Works.** Although the employment of spherical representations equipped with spherical attention mechanism yields superior performance, the computational complexity of the attention mechanism, which scales quadratically with the number of tokens, limits the resolution of spherical grids. When the sphere is evenly partitioned into $M = 768$ grids, each grid corresponds to an area of approximately 53.71 square degrees. For future work, we may explore linear attention mechanism (Wang et al., 2020) to facilitate spherical feature interaction. This could potentially enhance the resolution of spherical partitioning and improve the quality of spherical representations.

ACKNOWLEDGEMENT

This work was partially supported by the National Nature Science Foundation of China (Grant 62306294, 62071122), Basic Strengthening Program Laboratory Fund (No.NKLDSE2023A009), Dreams Foundation of Jianghuai Advance Technology Center (NO.2023M743385), Youth Innovation Promotion Association of CAS.

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

# A APPENDIX

## A.1 CLARIFICATIONS ON SO(3)-INVARIANCE

In this section, we first provide a detailed explanation of SO(3)-invariance and equivariance. Subsequently, we elaborate on the SO(3)-invariance of DINOv2 and ColorPointNet++, respectively.

**SO(3)-Invariance/Equivariance.** (1) *Rotation estimation network should be* SO(3)-*equivariant*. The rotation estimation task can be defined as $\boldsymbol{R} = \mathrm{Net}(\boldsymbol{P})$, where Net denotes the rotation estimation network. The SO(3)-equivariance between the input observed point cloud $\boldsymbol{P} \in \mathbb{R}^{N \times 3}$ and the output estimated rotation $\boldsymbol{R} \in \mathrm{SO}(3)$ is then formulated as:

$$\mathrm{Net}(\psi_{\boldsymbol{G}}(\boldsymbol{P})) = \psi_{\boldsymbol{G}}(\mathrm{Net}(\boldsymbol{P})). \tag{10}$$

(2) *Correspondence predictor should ensure* SO(3)-*invariance.* Correspondence-based methods can be expressed as $\boldsymbol{R} = \mathrm{Net}(\boldsymbol{P}) = \Phi(\boldsymbol{P}, \mathrm{Corr}(\boldsymbol{P}))$, where $\Phi$ is the rotation solver (*e.g.*, the Umeyama (Umeyama, 1991) algorithm) and Corr is the correspondence (NOCS) predictor. Then, SO(3)-equivariance between the network input and output is formulated as:

$$\Phi(\psi_{\boldsymbol{G}}(\boldsymbol{P}), \mathrm{Corr}(\psi_{\boldsymbol{G}}(\boldsymbol{P}))) = \psi_{\boldsymbol{G}}(\Phi(\boldsymbol{P}, \mathrm{Corr}(\boldsymbol{P}))). \tag{11}$$

Given fixed reference coordinates $\mathrm{Corr}(\boldsymbol{P}) \in \mathbb{R}^{N \times 3}$, the rotation solver $\Phi$ is SO(3)-equivariant with respect to the other input $\boldsymbol{P}$, which is written as:

$$\Phi(\psi_{\boldsymbol{G}}(\boldsymbol{P}), \mathrm{Corr}(\boldsymbol{P})) = \psi_{\boldsymbol{G}}(\Phi(\boldsymbol{P}, \mathrm{Corr}(\boldsymbol{P}))). \tag{12}$$

Consequently, the correspondence predictor Corr needs to be designed to ensure SO(3)-invariance:

$$\mathrm{Corr}(\psi_{\boldsymbol{G}}(\boldsymbol{P})) = \mathrm{Corr}(\boldsymbol{P}). \tag{13}$$

It means that no matter how an object is rotated, a specific point on that object should be mapped to a static coordinate in NOCS.

(3) *Feature extractor in correspondence-based methods should be* SO(3)-*invariant.* Since the correspondence predictor is typically defined as $\mathrm{Corr}(\boldsymbol{P}) = \mathrm{MLP}(f(\boldsymbol{P}))$, the point-wise feature extractor $f(\boldsymbol{P})$ in correspondence-based approaches also needs to be SO(3)-invariant, formulated as:

$$f(\psi_{\boldsymbol{G}}(\boldsymbol{P})) = f(\boldsymbol{P}), \tag{14}$$

which is consistent with Equation 1.

**SO(3)-invariance of DINOv2.** To illustrate the SO(3)-invariance of DINOv2 (Oquab et al., 2024), Figure 5 presents a PCA visualization of DINOv2 features from the same instance under different rotations. The visualization shows that the DINOv2 features of the camera lens remain relatively consistent across various rotations, indicating their robustness to rotation variations. However, since DINOv2 does not strictly guarantee such semantic consistency, we conclude the features as *approximately* SO(3)-invariant, in line with the terminology stated in Second-Pose (Chen et al., 2024).

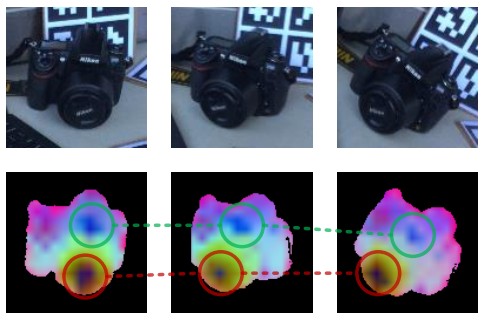

Figure 5: PCA visualization of DINOv2 features.

**SO(3)-invariance of ColorPointNet++.** To further illustrate the SO(3)-invariance of the proposed ColorPointNet++ network, we show the pipeline comparison between ColorPointNet++ and PointNet++ (Qi et al., 2017) in Figure 6. Given the input points $\boldsymbol{P} \in \mathbb{R}^{N \times d}$, where $d$ denotes the coordinate dimension and is equal to 3 in Euclidean space, and associated point attributes $\boldsymbol{F}^{attr} \in \mathbb{R}^{N \times C_0}$ (e.g., RGB colors with $C_0 = 3$), PointNet++ first concatenates them to yield the input features $\boldsymbol{F}^{in} \in \mathbb{R}^{N \times (d+C_0)}$, and then aggregates the point-wise features $\boldsymbol{F}^{out} \in \mathbb{R}^{N \times C}$ by a hierarchical grouping and propagation strategy. It should be noted that in each set abstraction level and feature propagation level, PointNet++ concatenates the absolute XYZ coordinates onto the output features. The injection of SO(3)-equivariant XYZ coordinates on features results in the output features of PointNet++ lacking SO(3)-invariance. In contrast, the proposed ColorPointNet++ is structured to

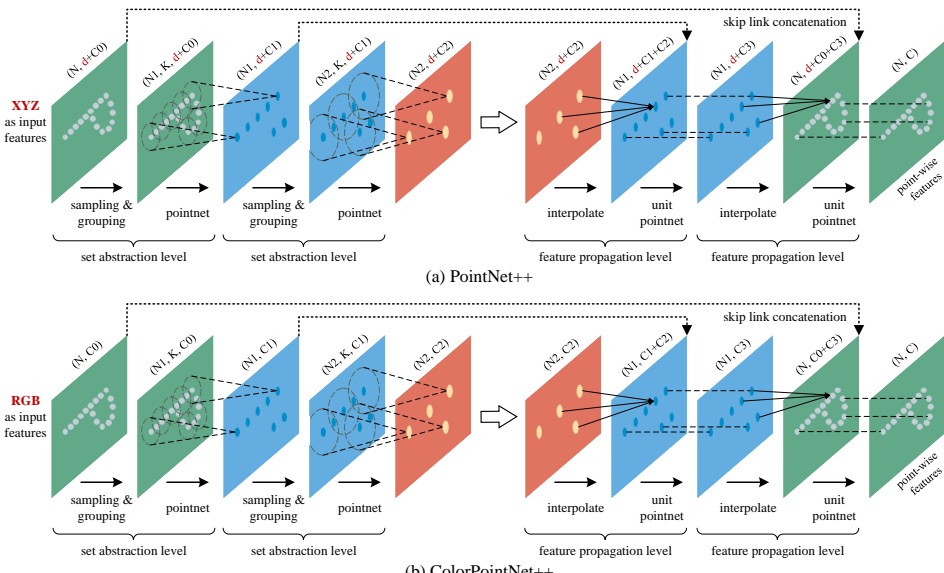

Figure 6: Overview of comparison between the PointNet++ and our ColorPointNet++ network.

achieve $SO(3)$-invariant feature extraction by avoiding the injection of absolute XYZ coordinates on the features. Specifically, XYZ coordinates are not used for input features and are not concatenated in each level, where the input features are derived exclusively from RGB values. Given that the k Nearest Neighbor (kNN) operation employed in the grouping and interpolation processes is $SO(3)$-invariant, the output features of our ColorPointNet++ network therefore exhibit $SO(3)$-invariance.

## A.2 MORE IMPLEMENTATION DETAILS

**Training Details.** For deep image feature extraction, we use the DINOv2 (Oquab et al., 2024) model with a ViT-S/14 (Dosovitskiy et al., 2020) architecture and feature dimension of 384. And for deep point cloud feature extraction, we implement the ColorPointNet++ network incorporating 4 set abstraction levels with multi-scale grouping, and the output feature dimension is 256. In terms of HEALPix grids (Gorski et al., 2005), the resolution is expressed by the parameter $N_{side}$ and the number of grids is equal to $N_{pix} = 12N_{side}^2$, where we set $N_{side} = 8$ and yield $M = N_{pix} = 768$. For data augmentation, we follow previous works (Lin et al., 2023b; 2024) to use perturbations with random translation $\Delta t \sim U(-0.02, 0.02)$, random scale $\Delta s \sim U(0.8, 1.2)$, and random rotational degree sampled from $U(0, 20)$ for each axis. Regarding network optimization, we train SpherePose using the Adam (Kingma & Ba, 2015) optimizer for 200K iterations, with an initial learning rate of 0.001 and a cosine annealing schedule.

**Correspondence Loss Functions.** The specific formulas corresponding to the loss functions in Table 6 within the ablation study section (Section 4.3) are provided below:

$$\mathcal{L}_{L^1} = \frac{1}{M} \sum_{m=1}^{M} \|\boldsymbol{O}_m - \boldsymbol{O}_m^{gt}\|_1, \tag{15}$$

$$\mathcal{L}_{\text{Smooth } L^1} = \frac{1}{M} \sum_{m=1}^{M} \begin{cases} 5\|\boldsymbol{O}_m - \boldsymbol{O}_m^{gt}\|_1^2, & \|\boldsymbol{O}_m - \boldsymbol{O}_m^{gt}\|_1 \leq 0.1 \\ \|\boldsymbol{O}_m - \boldsymbol{O}_m^{gt}\|_1 - 0.05, & \|\boldsymbol{O}_m - \boldsymbol{O}_m^{gt}\|_1 > 0.1 \end{cases}, \tag{16}$$

$$\mathcal{L}_{\text{Hyperbolic } L^1} = \frac{1}{M} \sum_{m=1}^{M} \text{arcosh}\left(1 + \|\boldsymbol{O}_m - \boldsymbol{O}_m^{gt}\|_1\right), \tag{17}$$

$$\mathcal{L}_{L^2} = \frac{1}{M} \sum_{m=1}^{M} \|\boldsymbol{O}_m - \boldsymbol{O}_m^{gt}\|_2, \tag{18}$$

$$\mathcal{L}_{\text{Hyperbolic } L^2} = \frac{1}{M} \sum_{m=1}^{M} \text{arcosh}\left(1 + \|\boldsymbol{O}_m - \boldsymbol{O}_m^{gt}\|_2\right), \tag{19}$$

Table 8: Performance comparison with state-of-the-art methods on CAMERA25 dataset.

| Method | | Representation | $IoU_{50}$ | $IoU_{75}$ | $5°2cm$ | $5°5cm$ | $10°2cm$ | $10°5cm$ |
|---|---|---|---|---|---|---|---|---|
| Direct Regression | GPV-Pose (Di et al., 2022) | Point-based | 93.4 | 88.3 | 72.1 | 79.1 | - | 89.0 |
| | HS-Pose (Zheng et al., 2023) | Point-based | 93.3 | 89.4 | 73.3 | 80.5 | 80.4 | 89.4 |
| | DualPoseNet (Lin et al., 2021) | Spherical | 92.4 | 86.4 | 64.7 | 70.7 | 77.2 | 84.7 |
| | VI-Net (Lin et al., 2023b) | Spherical | - | - | 74.1 | 81.4 | 79.3 | 87.3 |
| Correspondence | NOCS (Wang et al., 2019b) | Point-based | 83.9 | 69.5 | 32.3 | 40.9 | 48.2 | 64.4 |
| | SPD (Tian et al., 2020) | Point-based | 93.2 | 83.1 | 54.3 | 59.0 | 73.3 | 81.5 |
| | SGPA (Chen & Dou, 2021) | Point-based | 93.2 | 88.1 | 70.7 | 74.5 | 82.7 | 88.4 |
| | IST-Net (Liu et al., 2023b) | Point-based | 93.7 | 90.8 | 71.3 | 79.9 | 79.4 | 89.9 |
| | Query6DoF (Wang et al., 2023) | Point-based | 91.9 | 88.1 | 78.0 | 83.1 | 83.9 | 90.0 |
| | AG-Pose (Lin et al., 2024) | Point-based | 93.8 | 91.3 | 77.8 | 82.8 | **85.5** | 91.6 |
| | **SpherePose** | Spherical | **94.8** | **92.4** | **78.3** | **84.3** | 84.8 | **92.3** |

Table 9: IoU metric comparison with state-of-the-art methods on HouseCat6D dataset.

| Method | | Representation | $IoU_{25}$ / $IoU_{50}$ | Bottle | Box | Can | Cup | Remote | Teapot | Cutlery | Glass | Tube | Shoe |
|---|---|---|---|---|---|---|---|---|---|---|---|---|---|
| Direct Regression | FS-Net(Chen et al., 2021) | Point-based | 74.9 / 48.0 | 65.3 / 45.0 | 31.7 / 1.2 | 98.3 / 73.8 | 96.4 / 68.1 | 65.6 / 46.8 | 69.9 / 59.8 | 71.0 / 51.6 | 99.4 / 32.4 | 79.7 / 46.0 | 71.4 / 55.4 |
| | GPV-Pose(Di et al., 2022) | Point-based | 74.9 / 50.7 | 66.8 / 45.6 | 31.4 / 1.1 | 98.6 / 75.2 | 96.7 / 69.0 | 65.7 / 46.9 | 75.4 / 61.6 | 70.9 / 52.0 | 99.6 / 62.7 | 76.9 / 42.4 | 67.4 / 50.2 |
| | VI-Net(Lin et al., 2023b) | Spherical | 80.7 / 56.4 | 90.6 / 79.6 | 44.8 / 12.7 | **99.0** / 67.0 | 96.7 / 72.1 | 54.9 / 17.1 | 52.6 / 47.3 | 89.2 / **76.4** | 99.1 / 93.7 | **94.9** / 36.0 | 85.2 / 62.4 |
| | SecondPose(Chen et al., 2024) | Spherical | 83.7 / 66.1 | 94.5 / 79.8 | 54.5 / **23.7** | 98.5 / **93.2** | 99.8 / 82.9 | 53.6 / 35.4 | 81.0 / 71.0 | **93.5** / 74.4 | 99.3 / 92.5 | 75.6 / 35.6 | 86.9 / 73.0 |
| Correspondence | NOCS(Wang et al., 2019b) | Point-based | 50.0 / 21.2 | 41.9 / 5.0 | 43.3 / 6.5 | 81.9 / 62.4 | 68.8 / 2.0 | 81.8 / 59.8 | 24.3 / 0.1 | 14.7 / 6.0 | 95.4 / 49.6 | 21.0 / 4.6 | 26.4 / 16.5 |
| | AG-Pose(Lin et al., 2024) | Point-based | 81.8 / 62.5 | 82.3 / 62.8 | 57.2 / 7.7 | 97.1 / 83.6 | 97.9 / 79.6 | **87.0** / 66.2 | 63.4 / 60.9 | 77.2 / 62.0 | **100.0** / **99.4** | 83.4 / **53.4** | 72.0 / 50.0 |
| | **SpherePose** | Spherical | **88.8 / 72.2** | **98.6 / 87.4** | **58.9** / 5.6 | 97.4 / 87.2 | **100.0 / 97.7** | 79.0 / 63.5 | **94.8 / 87.3** | 85.5 / 74.5 | 99.6 / 98.0 | 74.5 / 30.0 | **99.8 / 90.8** |

## A.3 ADDITIONAL EXPERIMENTAL RESULTS

**Results on CAMERA25 Dataset.** In Table 8, we compare our method with the existing ones for category-level object pose estimation on CAMERA25 (Wang et al., 2019b) dataset. From the results we can see that SpherePose achieves the best performance. In detail, SpherePose outperforms the state-of-the-art correspondence-based method AG-Pose (Lin et al., 2024) by 1.0% on $IoU_{50}$, 1.1% on $IoU_{75}$, 0.5% on $5°2cm$ and 1.5% on $5°5cm$, respectively.

**Results of IoU Metric on HouseCat6D Dataset.** Table 9 presents the quantitative comparison of IoU metrics between our method and existing methods on HouseCat6D (Jung et al., 2024) dataset. Once again, our SpherePose attains the state-of-the-art performance, surpassing the second-best method SecondPose (Chen et al., 2024) by 5.1% on $IoU_{25}$ and 6.1% on $IoU_{50}$.

**Ablation Studies on Point-wise Features.** Table 10 presents the results of detailed ablation studies on point-wise features. For low-level image features, RGB values provide the low-level texture information, bringing a performance gain of 1.4% on $5°2cm$. For high-level image features, the frozen pretrained DINOv2 (Oquab et al., 2024) achieves a 4.4% higher performance on $5°2cm$ compared to

Table 10: Ablation studies on point-wise features.

| Point-wise Features | | $5°2cm$ | $5°5cm$ | $10°2cm$ | $10°5cm$ |
|---|---|---|---|---|---|
| Low-level Image | **RGB** | **58.2** | **67.4** | **76.2** | **88.2** |
| | None | 56.8 | 65.9 | 75.9 | 87.7 |
| High-level Image | **DINOv2** | **58.2** | **67.4** | **76.2** | **88.2** |
| | None | 48.7 | 57.1 | 69.2 | 80.3 |
| | ResNet18 | 53.8 | 62.5 | 72.8 | 83.8 |
| Low-level Point Cloud | **Radius** | **58.2** | **67.4** | **76.2** | **88.2** |
| | None | 52.3 | 58.9 | 72.3 | 83.6 |
| High-level Point Cloud | **ColorPointNet++** | **58.2** | **67.4** | **76.2** | **88.2** |
| | None | 55.9 | 64.6 | 74.3 | 86.7 |
| | PointNet++ | 56.3 | 64.7 | 73.2 | 84.4 |
| | HP-PPF | 55.1 | 64.2 | 74.2 | 85.9 |

the end-to-end trained ResNet18 (He et al., 2016). Due to the large-scale pretraining, DINOv2 produces semantically consistent patch-wise features that are robust to rotations, which facilitates correspondence prediction. For low-level point cloud features, removing the radius information results in a 5.9% drop on $5°2cm$, which highlights the importance of spatial structure awareness. Since the proposed ColorPointNet++ eliminates the injection of raw XYZ coordinates, its features lack implicit encoding of the underlying spatial structure, making explicit SO(3)-invariant radius information essential. For high-level point cloud features, ColorPointNet++ outperforms the original PointNet++ (Qi et al., 2017) by 3.8% on $10°5cm$, emphasizing the significance of SO(3)-invariant features. Additionally, We have tried HP-PPF (Chen et al., 2024), which is also SO(3)-invariant, but the performance is suboptimal compared to not using it. This might stem from the fact that HP-PPF samples 300 points from 2048 observed points to estimate point-wise normal vectors, making it prone to noise due to the sparse sampling. Moreover, as a parameter-free approach, HP-PPF lacks the deep feature extraction capabilities compared to the learnable architecture of ColorPointNet++.

Table 11: Per-category results of our Sphere-Pose on REAL275 dataset.

| Category | IoU$_{50}$ | IoU$_{75}$ | 5°2cm | 5°5cm | 10°2cm | 10°5cm |
|---|---|---|---|---|---|---|
| bottle | 57.6 | 51.3 | 71.7 | 76.8 | 81.4 | 88.7 |
| bowl | 100.0 | 100.0 | 87.7 | 96.9 | 90.8 | 100.0 |
| camera | 90.9 | 77.5 | 5.4 | 5.7 | 43.9 | 50.1 |
| can | 71.4 | 70.6 | 76.5 | 81.6 | 91.9 | 98.4 |
| laptop | 84.8 | 75.5 | 59.5 | 92.7 | 60.0 | 97.0 |
| mug | 99.6 | 99.3 | 48.5 | 50.3 | 89.2 | 94.9 |
| average | 84.0 | 79.0 | 58.2 | 67.4 | 76.2 | 88.2 |

Table 12: Per-category results of AG-Pose (Lin et al., 2024) on REAL275 dataset.

| Category | IoU$_{50}$ | IoU$_{75}$ | 5°2cm | 5°5cm | 10°2cm | 10°5cm |
|---|---|---|---|---|---|---|
| bottle | 57.7 | 50.3 | 62.0 | 64.9 | 83.4 | 88.0 |
| bowl | 100.0 | 100.0 | 88.7 | 94.3 | 94.1 | 99.7 |
| camera | 90.8 | 82.9 | 1.2 | 1.3 | 24.8 | 27.3 |
| can | 71.3 | 71.2 | 83.4 | 85.3 | 96.3 | 98.6 |
| laptop | 83.3 | 74.1 | 59.6 | 91.1 | 61.2 | 95.6 |
| mug | 99.4 | 98.5 | 32.9 | 33.4 | 88.3 | 89.3 |
| average | 83.7 | 79.5 | 54.7 | 61.7 | 74.7 | 83.1 |

Table 13: Per-category results of our Sphere-Pose on CAMERA25 dataset.

| Category | IoU$_{50}$ | IoU$_{75}$ | 5°2cm | 5°5cm | 10°2cm | 10°5cm |
|---|---|---|---|---|---|---|
| bottle | 93.8 | 90.6 | 78.0 | 93.3 | 79.5 | 96.4 |
| bowl | 97.0 | 96.7 | 95.1 | 95.6 | 98.6 | 99.2 |
| camera | 94.5 | 91.6 | 69.2 | 72.8 | 78.9 | 86.9 |
| can | 92.2 | 91.7 | 96.7 | 97.9 | 97.2 | 98.6 |
| laptop | 97.9 | 91.4 | 73.1 | 87.4 | 75.6 | 92.1 |
| mug | 93.6 | 92.3 | 57.6 | 58.9 | 78.7 | 80.4 |
| average | 94.8 | 92.4 | 78.3 | 84.3 | 84.8 | 92.3 |

Table 14: Per-category results of AG-Pose (Lin et al., 2024) on CAMERA25 dataset.

| Category | IoU$_{50}$ | IoU$_{75}$ | 5°2cm | 5°5cm | 10°2cm | 10°5cm |
|---|---|---|---|---|---|---|
| bottle | 93.7 | 91.4 | 80.9 | 96.4 | 82.3 | 99.0 |
| bowl | 96.9 | 96.7 | 98.7 | 99.0 | 99.7 | 99.8 |
| camera | 89.2 | 84.3 | 57.0 | 60.9 | 73.6 | 81.1 |
| can | 92.1 | 92.0 | 99.7 | 99.8 | 99.7 | 99.9 |
| laptop | 97.5 | 90.8 | 76.1 | 85.9 | 80.6 | 92.4 |
| mug | 93.6 | 92.7 | 54.5 | 54.6 | 77.1 | 77.3 |
| average | 93.8 | 91.3 | 77.8 | 82.8 | 85.5 | 91.6 |

**Per-category Results.** The REAL275 and CAMERA25 datasets share the same categories, including bottle, bowl, camera, can, laptop and mug. In order to conduct an in-depth analysis of the relative merits of our method, we show the per-category and average results of our Sphere-Pose and AG-Pose (Lin et al., 2024) on REAL275 dataset in Table 11 and Table 12, and on CAMERA dataset in Table 13 and Table 14, respectively. It can be observed that SpherePose outperforms AG-Pose mainly in the **camera** and **mug** categories, which have large intra-class shape variations. This result demonstrates the effective-

Table 15: Per-category results of our Sphere-Pose on HouseCat6D dataset.

| Category | IoU$_{25}$ | IoU$_{50}$ | 5°2cm | 5°5cm | 10°2cm | 10°5cm |
|---|---|---|---|---|---|---|
| bottle | 98.6 | 87.4 | 51.0 | 55.7 | 67.9 | 79.1 |
| box | 58.9 | 5.6 | 0.0 | 0.0 | 0.2 | 0.2 |
| can | 97.4 | 87.2 | 33.9 | 41.4 | 63.0 | 79.2 |
| cup | 100.0 | 97.7 | 2.8 | 3.0 | 60.6 | 68.4 |
| remote | 79.0 | 63.5 | 15.7 | 16.3 | 46.9 | 50.1 |
| teapot | 94.8 | 87.3 | 25.2 | 44.4 | 53.6 | 86.8 |
| cutlery | 85.5 | 74.5 | 1.4 | 1.6 | 8.2 | 13.0 |
| glass | 99.6 | 98.0 | 54.8 | 66.3 | 78.5 | 96.4 |
| tube | 74.5 | 30.0 | 1.0 | 1.2 | 5.3 | 6.7 |
| shoe | 99.8 | 90.8 | 7.3 | 29.3 | 24.5 | 73.2 |
| average | 88.8 | 72.2 | 19.3 | 25.9 | 40.9 | 55.3 |

ness of our approach to learn shape-independent transformation via spherical representations, which can resolve the issue of intra-class semantic incoherence arising from point-based representations. We also provide the per-category results on HouseCat6D dataset in Table 15, which contains 10 household categories: bottle, box, can, cup, remote, teapot, cutlery, glass, tube and shoe. Note that we deal with all categories by a single model as in Lin et al. (2022b; 2023b; 2024).

**Convergence Speed of the Hyperbolic Correspondence Loss Function.** Figure 7 visualizes the training convergence speed and accuracy comparison between our hyperbolic correspondence loss function and the traditional loss function. As discussed in Section 3.3, the hyperbolic $L^2$ loss function is designed to improve the gradient of the traditional $L^2$ loss function around zero, facilitating more fine-grained correspondence prediction. As shown in the figure, the hyperbolic $L^2$ loss not only accelerates convergence during training but also achieves higher accuracy, demonstrating its effectiveness in improving correspondence prediction precision.

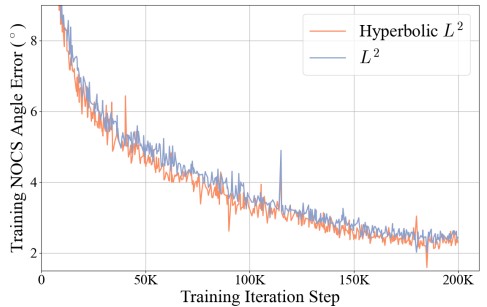

Figure 7: Training comparison between the hyperbolic $L^2$ and traditional $L^2$ loss functions.

**Inference Speed and Overhead.** Table 16 shows the comparison of the pose estimation performance, total number of parameters, and inference speed on REAL275 dataset. As demonstrated by the results, the increase in the number $L$ of encoder layers leads to enhanced performance, reaching saturation at six layers. A further increase in the number of layers does not result in further perfor-

Table 16: Comparison of different methods in terms of performance on REAL275, inference speed and overhead. All the experiments for inference speed are conducted on a single RTX3090Ti GPU.

| Method | $5°2cm$ | $5°5cm$ | $10°2cm$ | $10°5cm$ | Parameters (M) | Speed (FPS) |
|---|---|---|---|---|---|---|
| VI-Net (Lin et al., 2023b) | 50.0 | 57.6 | 70.8 | 82.1 | 28.9 | 29.7 |
| SecondPose (Chen et al., 2024) | 56.2 | 63.6 | 74.7 | 86.0 | 60.2 | 14.3 |
| DPDN (Lin et al., 2022b) | 46.0 | 50.7 | 70.4 | 78.4 | 24.6 | 27.4 |
| AG-Pose (Lin et al., 2024) | 54.7 | 61.7 | 74.7 | 83.1 | 28.9 | 27.5 |
| SpherePose with $L = 2$ | 53.6 | 61.3 | 74.0 | 85.8 | 25.7 | 26.4 |
| SpherePose with $L = 4$ | 55.7 | 64.5 | 75.2 | 86.5 | 26.1 | 25.9 |
| SpherePose with $L = 6$ | **58.2** | **67.4** | **76.2** | **88.2** | 26.5 | 25.3 |
| SpherePose with $L = 8$ | 57.9 | 66.7 | 74.9 | 86.2 | 26.9 | 24.5 |

mance gains, but rather increases the number of parameters and slows down the inference. Therefore, we set $L = 6$ in our method. Moreover, our approach demonstrates superior performance over existing methods, retaining a comparable number of parameters and inference speed. Specifically, in comparison to the leading correspondence-based method AG-Pose (Lin et al., 2024), our Sphere-Pose achieves a 3.5% improvement on $5°2cm$, with 2.4 million fewer parameters and only a slight decrease in FPS. When compared to the direct regression-based methods, SpherePose outperforms the advanced SecondPose (Chen et al., 2024) by 2% on $5°2cm$, with substantially fewer parameters (26.5 million and 60.2 million, respectively) and higher FPS (25.3 and 14.3, respectively).

## A.4 CONCURRENT RELATED WORKS

In this section, we discuss the distinctions between our approach and two concurrent related works.

**Correspondence-based work.** Mariotti et al. (2024) propose a semantic correspondence estimation approach, which deal with the challenges of object symmetries and repeated parts by incorporating a weak geometric spherical prior to supplement leading self-supervised features with 3D perception capabilities. They project image features onto a spherical map and use viewpoint information to guide correspondence prediction. While both this work and ours utilize spherical representations, the primary focus and application are different. Mariotti et al. (2024) aim to improve semantic correspondence estimation, which involves finding local regions that correspond to the same semantic entities across images. In contrast, our SpherePose focuses on correspondence-based category-level object pose estimation, which involves establishing correspondences between observed points and normalized object coordinates, and then fitting the object pose. Mariotti et al. (2024) use spherical representations to learn 3D perception-driven semantic features, while we leverage spherical representations to learn shape-independent transformation, which is beneficial for handling large intra-class shape variations.

**Regression-based work.** Lee & Cho (2024) propose a 3D rotation regression approach that directly predicts Wigner-D coefficients in the frequency domain, aligning with the operations of spherical CNNs. They uses an $SO(3)$-equivariant pose harmonics predictor to ensure consistent pose estimation under arbitrary rotations, overcoming limitations of spatial parameterizations such as discontinuities and singularities. While both this work and ours aim to improve 3D rotation estimation, the operating domains and the way to estimate rotation are different. Lee & Cho (2024) focus on predicting Wigner-D coefficients in the frequency domain to capture 3D rotations, which is useful for ensuring consistent pose estimation under different rotations. Instead, our method operates in the spatial domain, leveraging spherical representations to learn shape-independent transformation rather than directly predicting rotations through the network.

## A.5 VISUALIZATION

**Qualitative Comparison.** In Figure 8, we provide the qualitative comparison of existing correspondence-based methods, i.e., DPDN (Lin et al., 2022b), AG-Pose (Lin et al., 2024) and our SpherePose on REAL275 dataset. The visualization results indicate that previous methods adopting point-based representations do not perform well when dealing with novel objects with large shape variations, such as the length variation in camera lenses and the curvature distinctions in mug han-

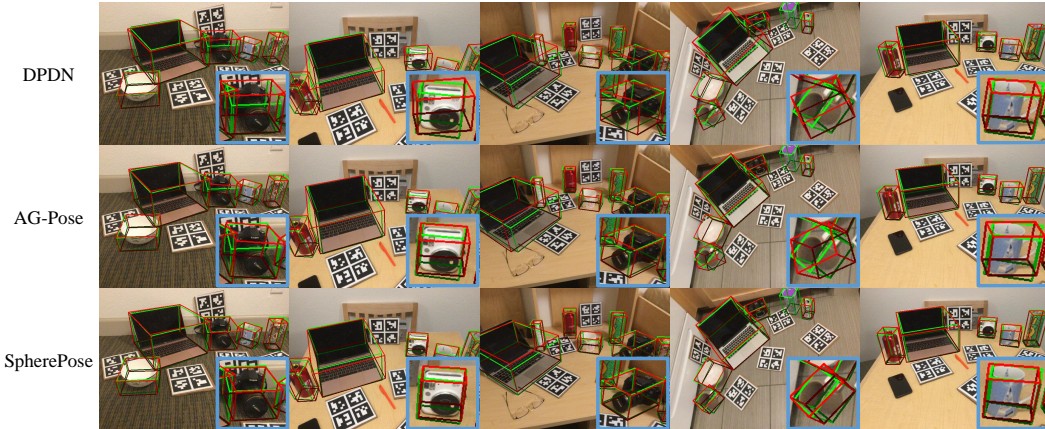

Figure 8: Qualitative comparison of DPDN (Lin et al., 2022b), AG-Pose (Lin et al., 2024) and our SpherePose on REAL275 dataset. Red/Green indicates the predicted/GT results.

dles. By learning shape-independent transformation on the sphere, the proposed method allows for enhanced pose estimation.

**More Visualization.** We provide more visualization results of pose estimation from our SpherePose, including successful cases in Figure 9 and failed cases in Figure 10. We visualize all six previously unseen scenes within the REAL275 testing dataset, with three images for each scene. It can be observed that failures predominantly occur due to the omission of objects in the detection process and significant viewpoint variations that result in incomplete capture of some objects.

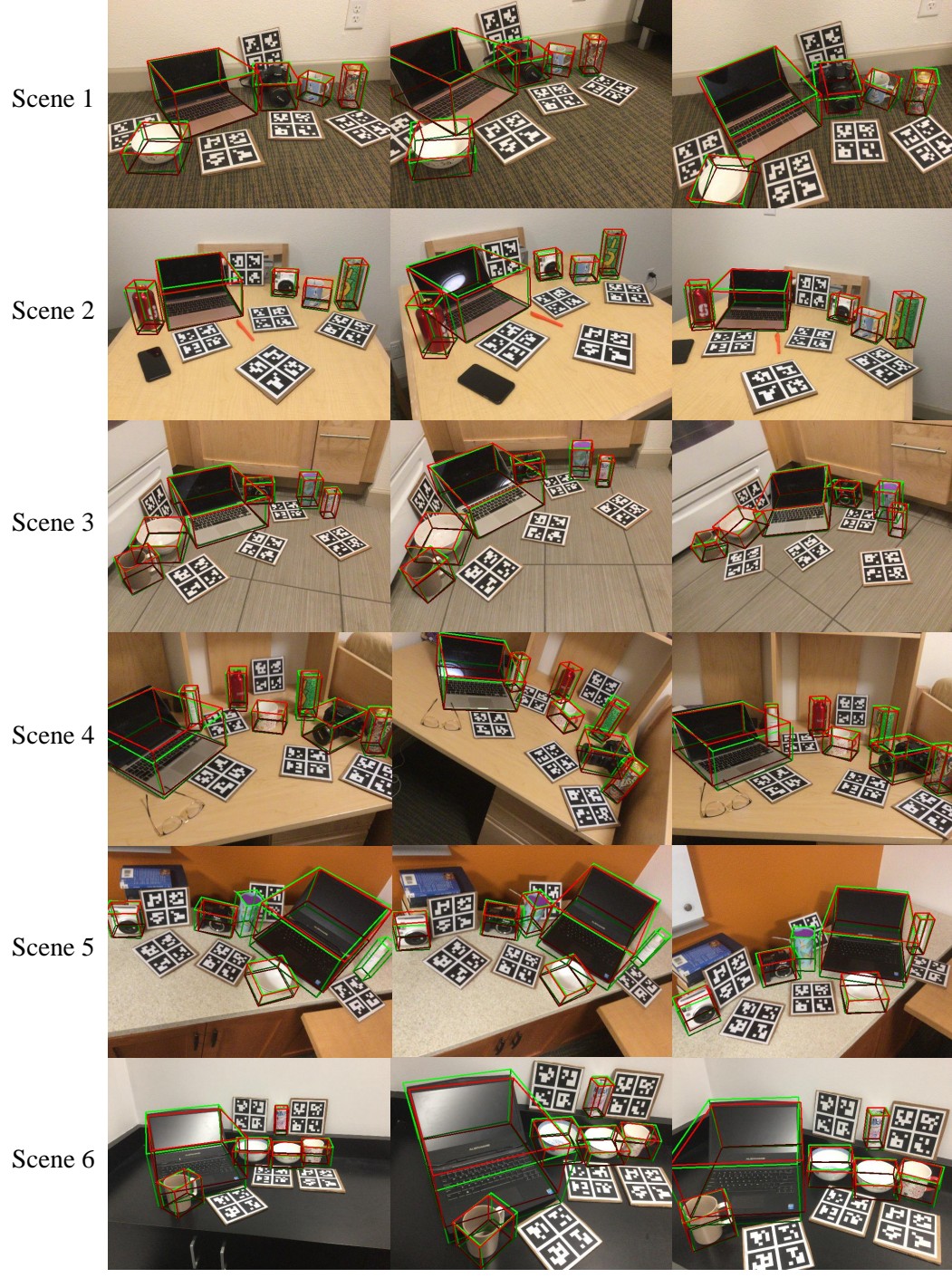

Figure 9: Visualization of successful cases from our SpherePose on the 6 testing scenes of REAL275 dataset. Red/Green indicates the predicted/GT results.

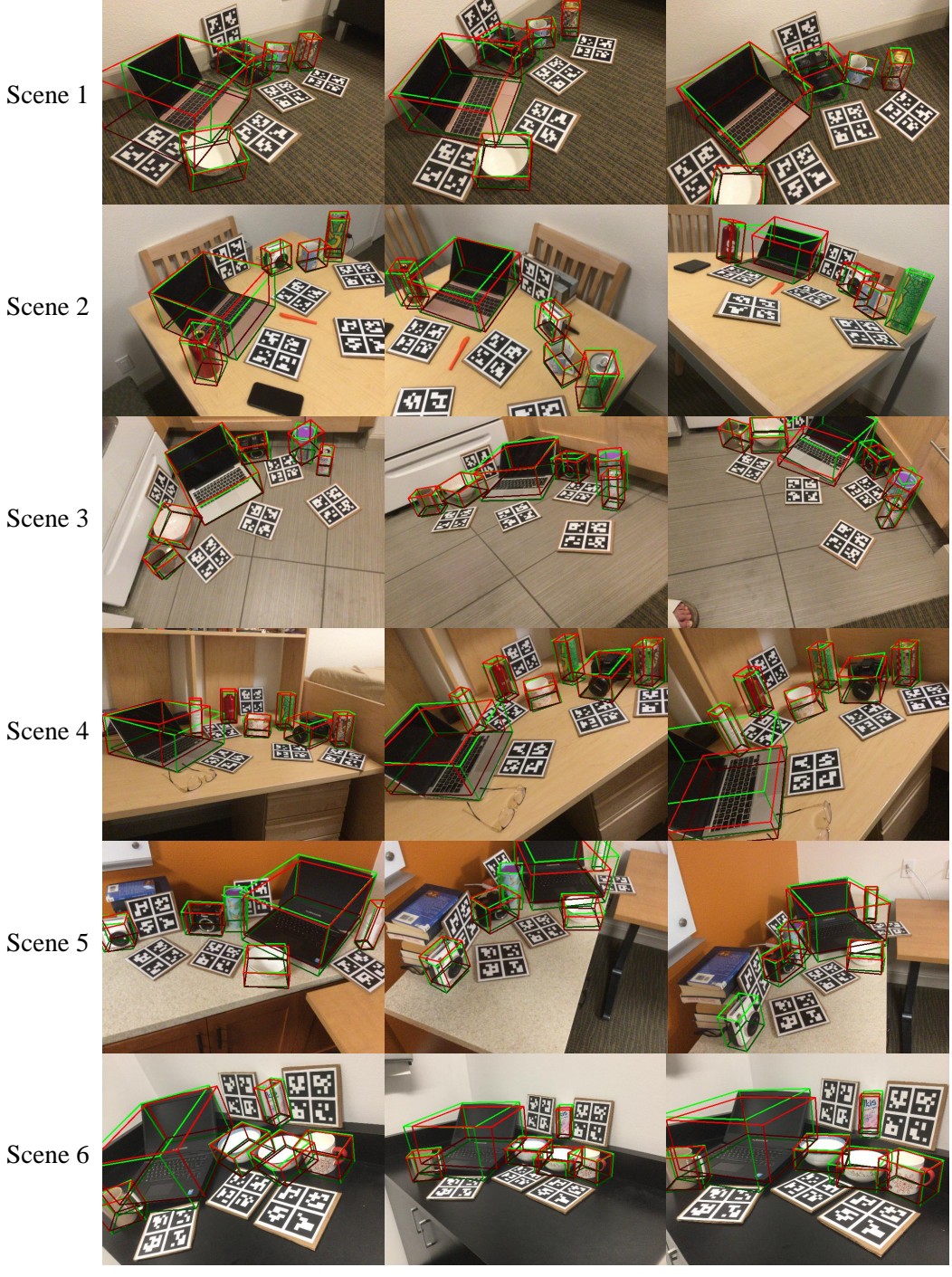

Figure 10: Visualization of failed cases from our SpherePose on the 6 testing scenes of REAL275 dataset. Red/Green indicates the predicted/GT results.

