# OpenReview forum: "Learning Shape-Independent Transformation via Spherical Representations for Category-Level Object Pose Estimation"
_ICLR.cc/2025/Conference — ICLR 2025 Poster_

### Official Review · Reviewer_p4nq · 2024-10-30

**Soundness:** 3
**Presentation:** 3
**Contribution:** 3
**Rating:** 6
**Confidence:** 4

**Summary:**

This paper introduces a new method for category-level object pose estimation. The authors argue that the shape-dependent representations in previous approaches lead to semantic inconsistencies across different objects within the same category. To handle this issue,  a shape-independent transformation is presented, transforming features to a sphere. The authors also introduce the importance of SO(3)-invariant features for category-level object pose estimation. Multiple SO(3)-invariant features extracted from RGB images and point clouds are combined to learn the NOCS coordinates on a sphere. Experimental results show SpherePose outperforms state-of-the-art methods on benchmarks like CAMERA25, REAL275, and HouseCat6D, showing its effectiveness in handling category-level pose estimation challenges.

**Strengths:**

The idea of leveraging an object-agnostic representation to handle the problem of semantic inconsistencies is technically sound and interesting. The authors propose to combine multiple deep feature extractors, and its effectiveness is demonstrated in the ablation studies. The paper conducts extensive experiments on multiple benchmarks, showing better performance compared with existing methods. The paper is well-written and easy to understand.

**Weaknesses:**

-	The presented ColorPointNet++ is not impressive. There are many alternatives [1,2,3] in the literature which can be used to extract SO(3)-invariant features.

-	The effectiveness of DINOv2 towards rotation invariance is questionable. The authors mentioned that it is “approximately” SO(3)-invariant, which is quite vague. It would be more convincing if the authors could provide some experimental results about the invariance of DINOv2 towards 3D rotations.

-	The difference compared with some previous methods that also use sphere representations, such as SpherePose, is unclear.  What is the major superiority compared with these methods?

-	Some important ablation studies are missing. For instance, the authors present a spherical feature interaction module to handle the challenges of self-occlusion. However, an ablation study on the effectiveness of this module is missing. Moreover, an experiment regarding the importance of SO(3)-invariant features for category-level object pose estimation is missing.

[1] Deng, Congyue, et al. "Vector neurons: A general framework for so (3)-equivariant networks." Proceedings of the IEEE/CVF International Conference on Computer Vision. 2021.

[2] Zhao, Chen, et al. "Rotation invariant point cloud analysis: Where local geometry meets global topology." Pattern Recognition 127 (2022): 108626.

[3] Fei, Jiajun, and Zhidong Deng. "Rotation invariance and equivariance in 3D deep learning: a survey." Artificial Intelligence Review 57.7 (2024): 168.

**Questions:**

-	I am a bit confused about the SO(3)-invariant features in the field of object rotation estimation. We could formulate the rotation estimation problem as R = f(g(x)|w). g(x) means the feature extractor. If the extracted features are rotation-invariant, it would be g(R(x))=g(x). If this is the case, it means f(.) would be unaware of the rotation information. Why is the network able to make the prediction related to rotations? To me, SO(3)-equivariant features seem more reasonable.
-	The proposed spherical representation is designed to be object-agnostic across different categories. What factors limit the method’s ability to generalize to objects from novel categories?
-	Why is the IoU75 worse than that of a point-based method AG-Pose on REAL275?

---

> ### Author Response · Authors · 2024-11-24
> **Response to Reviewer p4nq (Part 1)**
>
> We appreciate the reviewer for the constructive comments of our work. We hope our following responses can address your concerns.
>
> ------
>
> **W1. ColorPointNet++ is not Impressive**
>
> > The presented ColorPointNet++ is not impressive. There are many alternatives [1,2,3] in the literature which can be used to extract SO(3)-invariant features.
> >
> > [1] Deng, Congyue, et al. "Vector neurons: A general framework for so (3)-equivariant networks." Proceedings of the IEEE/CVF International Conference on Computer Vision. 2021.
> >
> > [2] Zhao, Chen, et al. "Rotation invariant point cloud analysis: Where local geometry meets global topology." Pattern Recognition 127 (2022): 108626.
> >
> > [3] Fei, Jiajun, and Zhidong Deng. "Rotation invariance and equivariance in 3D deep learning: a survey." Artificial Intelligence Review 57.7 (2024): 168.
>
> * **This work aims to validate the importance of SO(3)-invariance for correspondence prediction, rather than focusing on specific network architectures.** To ensure fair comparisons with prior works, we maintain the widely used PointNet++ architecture and parameter count, making minimal adjustments to transform it from SO(3)-sensitive to SO(3)-invariant. The ablation studies in Table 4 demonstrate the effectiveness of this insight.
>
> * **Other SO(3)-invariant point cloud backbones could also be adopted.** We leave this exploration for future work.
>
> ------
>
> **W2. Rotation Invariance of DINOv2**
>
> > The effectiveness of DINOv2 towards rotation invariance is questionable. The authors mentioned that it is “approximately” SO(3)-invariant, which is quite vague. It would be more convincing if the authors could provide some experimental results about the invariance of DINOv2 towards 3D rotations.
>
> Thanks for the suggestion. Please refer to **Figure 6** in the **Appendix** of our revised manuscript for more details.
>
> * **We have included an analysis of the SO(3)-invariance of DINOv2 [7] features in Figure 6 in the Appendix.**
> * **SecondPose [11] also points out that DINOv2 features are approximately SO(3)-invariant**. We adopt the same description in our work.
>
> ------
>
> **W3. Comparison with Methods based on Sphere Representations**
>
> > The difference compared with some previous methods that also use sphere representations, such as SpherePose, is unclear. What is the major superiority compared with these methods?
>
> * **We have already discussed the differences and advantages of our approach compared to existing category-level object pose estimation methods using spherical representations in Section 3.4.** For more details, please refer to Section 3.4 in the main text.
> * **SpherePose refers to our proposed method.** We are unsure which specific work you are asking us to compare against. Please let us know if you have more questions.
>
> ------
>
> **W4. Missing Ablation Studies**
>
> > Some important ablation studies are missing. For instance, the authors present a spherical feature interaction module to handle the challenges of self-occlusion. However, an ablation study on the effectiveness of this module is missing. Moreover, an experiment regarding the importance of SO(3)-invariant features for category-level object pose estimation is missing.
>
> * **The effectiveness of spherical feature interaction is demonstrated in Table 5.** When the number of encoder layers is set to 0, indicating no spherical feature interaction is used, the network struggles to handle occluded anchors effectively, resulting in a significant performance drop (from 58.2% to 20.7% in 5°2cm). This highlights the importance of spherical feature interaction. For more detailed discussions on how it aids in reasoning about the features of occluded anchors, please refer to the common response **R3** above.
> * **The effectiveness of SO(3)-invariant features is validated in Table 4.** The original PointNet++ lacks SO(3)-invariance, while our ColorPointNet++ is designed to be SO(3)-invariant. With comparable parameter counts and architecture, ColorPointNet++ achieves 1.9% higher performance in 5°2cm and 3.8% in 10°5cm. These results demonstrate the importance of SO(3)-invariant features.

---

> ### Author Response · Authors · 2024-11-24
> **Response to Reviewer p4nq (Part 2)**
>
> **Q1. Explanation about SO(3)-Invariance/Equivariance**
>
> > I am a bit confused about the SO(3)-invariant features in the field of object rotation estimation. We could formulate the rotation estimation problem as R = f(g(x)|w). g(x) means the feature extractor. If the extracted features are rotation-invariant, it would be g(R(x))=g(x). If this is the case, it means f(.) would be unaware of the rotation information. Why is the network able to make the prediction related to rotations? To me, SO(3)-equivariant features seem more reasonable.
>
> Please refer to our common response **R1** above for more details.
>
> * **Rotation estimation network must maintain SO(3)-equivariance between its input and output.**
> * **Correspondence predictor in correspondence-based methods must ensure SO(3)-invariance.**
> * **Features extracted in correspondence-based methods must ensure SO(3)-invariance.**
>
> ------
>
> **Q2. Generalization to Novel Categories**
>
> > The proposed spherical representation is designed to be object-agnostic across different categories. What factors limit the method’s ability to generalize to objects from novel categories?
>
> * **Definition of the canonical poses for novel categories and identification for new objects.** The key factor limiting generalization to novel categories in category-level object pose estimation task is the definition of their canonical poses, as the standard poses for new categories is unknown. Additionally, the ability to detect and segment objects from novel categories also affects the generalization capability.
>
> ------
>
> **Q3. IoU75 Performance Comparison**
>
> > Why is the IoU75 worse than that of a point-based method AG-Pose on REAL275?
>
> * **This work focuses on rotation estimation, while the IoU metric is jointly influenced by rotation, translation, and size estimation.** Following VI-Net [2], our primary focus is on the more challenging task of rotation estimation, while translation and size are estimated using a lightweight PointNet++, which may lead to slightly lower performance in translation and size.
>
> ------
>
> **References**
>
> [2] Lin J, Wei Z, Zhang Y, et al. Vi-net: Boosting category-level 6d object pose estimation via learning decoupled rotations on the spherical representations[C]//Proceedings of the IEEE/CVF International Conference on Computer Vision. 2023: 14001-14011.
>
> [7] Oquab M, Darcet T, Moutakanni T, et al. DINOv2: Learning Robust Visual Features without Supervision[J]. Transactions on Machine Learning Research.
>
> [11] Chen Y, Di Y, Zhai G, et al. Secondpose: Se (3)-consistent dual-stream feature fusion for category-level pose estimation[C]//Proceedings of the IEEE/CVF Conference on Computer Vision and Pattern Recognition. 2024: 9959-9969.

---

> > ### Comment · Reviewer_p4nq · 2024-11-25
> >
> > I thank the authors for the rebuttal. My concerns about invariance and equivariance have been addressed, so I am glad to keep my positive rating. Since the authors also agree that this paper does not focus on network architectures, it is better to rewrite the part about ColorPointNet++, which is not an impressive contribution to me. Moreover, I still cannot entirely agree the DINOv2 feature is rotation-invariant. It might be better to claim that it is robust to rotations.

---

> > > ### Author Response · Authors · 2024-11-27
> > > **Response to Further Suggestions by Reviewer p4nq**
> > >
> > > We sincerely thank you for your constructive suggestions. In the revised manuscript, we have rewritten the part about ColorPointNet++, emphasizing the importance of SO(3)-invariance rather than focusing on the specifics of the network architecture. Additionally, we have refined the description of DINOv2 features, referring to them as "robust to rotations" rather than rotation-invariant. If you have any further questions or suggestions, we would be grateful for your comments to help us improve our work.

---

### Official Review · Reviewer_v4xt · 2024-11-02

**Soundness:** 3
**Presentation:** 3
**Contribution:** 3
**Rating:** 6
**Confidence:** 4

**Summary:**

This paper addresses the task of category-level object pose and size estimation, introducing a novel method called SpherePose. This method uses a sphere as a shared proxy shape for objects, enabling the learning of shape-independent transformations from spherical representations. To enhance the precision of correspondences on the sphere, SpherePose incorporates three core components, including  SO(3)-invariant point-wise feature extraction, spherical feature interaction, and a hyperbolic correspondence loss function. Experiments conducted on the CAMERA25, REAL275, and HouseCat6D datasets validate the effectiveness of SpherePose.

**Strengths:**

- Unlike point-based representations, SpherePose uses a sphere as a shared proxy shape for objects and employs spherical representations to learn shape-independent transformations.

- Three core components are introduced based on spherical representations to enhance the precision of correspondences.

- SpherePose achieves state-of-the-art results on the CAMERA25, REAL275, and HouseCat6D datasets.

**Weaknesses:**

- Are the spherical NOCS coordinates derived by normalizing the original NOCS coordinates to unit vectors? If so, it would be beneficial to provide results for regressing the original NOCS coordinates.

- Are the resulting poses obtained from the observed anchors or all sampled anchors on the sphere? If it’s the former, how can we verify that the spherical feature interaction aids in reasoning about the features of occluded anchors?

- Unlike most existing methods that use the Umeyama algorithm, SpherePose incorporates RANSAC for solving rotations. For a fair comparison, results without RANSAC for SpherePose should be included in Table 1.

- It is recommended to include results that do not use RGB or radius values as inputs in Table 4.

**Questions:**

See Weaknesses.

---

> ### Author Response · Authors · 2024-11-24
> **Response to Reviewer v4xt**
>
> We appreciate the reviewer for the constructive comments of our work. We hope our following responses can address your concerns.
>
> ------
>
> **W1. Derivation of Spherical NOCS Coordinates**
>
> > Are the spherical NOCS coordinates derived by normalizing the original NOCS coordinates to unit vectors? If so, it would be beneficial to provide results for regressing the original NOCS coordinates.
>
> Please refer to our common response **R4** above for more details.
>
> * **Spherical NOCS coordinates are inherently unit vectors.**
>
> ------
>
> **W2. Pose Determination from Anchors**
>
> > Are the resulting poses obtained from the observed anchors or all sampled anchors on the sphere? If it’s the former, how can we verify that the spherical feature interaction aids in reasoning about the features of occluded anchors?
>
> Please refer to our common response **R3** above for more details.
>
> * **Spherical feature interaction aids in reasoning about the features of occluded anchors.** In our method, correspondences from all anchors are used by default for rotation fitting. Through an in-depth analysis, we observe that even using correspondences from only occluded anchors achieves comparable fitting performance. This result demonstrates the capability of spherical feature interaction to effectively reason about the features of occluded anchors.
>
> ------
>
> **W3. Umeyama without RANSAC**
>
> > Unlike most existing methods that use the Umeyama algorithm, SpherePose incorporates RANSAC for solving rotations. For a fair comparison, results without RANSAC for SpherePose should be included in Table 1.
>
> We are sorry for the potential confusion:
>
> - **Existing methods using the Umeyama algorithm typically rely on RANSAC for outlier removal.** For example, the pioneering work on category-level object pose estimation, NOCS [5], employs the Umeyama algorithm with RANSAC for pose fitting. Subsequent works that adopt the Umeyama algorithm have also used RANSAC by default for outlier removal.
> - **RANSAC is not essential for our SpherePose.** The spherical feature interaction, conducted from a holistic perspective, ensures consistent correspondence predictions among spherical anchors, mitigating the interference of noise. As shown in the table below, SpherePose achieves comparable performance even without RANSAC.
>
> | Umeyama    | 5°2cm | 5°5cm | 10°2cm | 10°5cm |
> | ---------- | ----- | ----- | ------ | ------ |
> | w/ RANSAC  | 58.2  | 67.4  | 76.2   | 88.2   |
> | w/o RANSAC | 58.2  | 67.4  | 76.2   | 88.2   |
>
> ------
>
> **W4. Additional Results in Table 4**
>
> > It is recommended to include results that do not use RGB or radius values as inputs in Table 4.
>
> Thanks for the suggestion. We have included results that exclude RGB or radius values. Please refer to **Table 11** in the **Appendix** of our revised manuscript for more details.
>
> ------
>
> **References**
>
> [5] Wang H, Sridhar S, Huang J, et al. Normalized object coordinate space for category-level 6d object pose and size estimation[C]//Proceedings of the IEEE/CVF Conference on Computer Vision and Pattern Recognition. 2019: 2642-2651.

---

> ### Author Response · Authors · 2024-11-27
> **Looking Forward to Feedback**
>
> We sincerely thank you for your time in reviewing our paper and your constructive comments. We have posted our point-to-point responses in the review system. Since the public discussion phase will end soon, we appreciate if you could read our responses and let us know your feedback and further comments.

---

### Official Review · Reviewer_rB6i · 2024-11-04

**Soundness:** 3
**Presentation:** 2
**Contribution:** 3
**Rating:** 6
**Confidence:** 4

**Summary:**

This paper addresses the challenge of category-level object pose estimation by proposing a shape-independent representation using spherical features. Starting with an RGB-D image and instance masks, the method processes "N" partial point clouds to estimate object pose parameters: rotation R \in SO(3), translation t \in R^3, and size s \in R^3 of the observed instance. The spherical feature construction on the SO(3) HEALPix grid, combined with a hyperbolic correspondence loss, significantly improves pose estimation performance. The paper categorizes pose estimation into correspondence-based vs. regression-based and point-based vs. spherical-based approaches. The proposed method, correspondence-based/spherical-based SpherePose, demonstrates state-of-the-art results on REAL275 and HouseCat6D benchmarks.

**Strengths:**

1. Strong motivation: The proposed shape-independent proxy representation addresses a crucial need in category-level pose estimation, enhancing generalizability across different object instances.

2. Good analysis of correspondence errors: The method effectively identifies correspondences, as shown in Table 3, where accurate correspondence significantly enhances pose estimation accuracy.
However, consistency of correspondence errors should be validated on another dataset, such as HouseCat6D.

3. Comprehensive Ablation Studies: 	The authors provide a detailed analysis of different feature extractors (DINOv2, ColorPointNet++) and loss functions (L2 vs. hyperbolic L2). While not the primary contribution, Table 4 highlights the role of backbone networks in performance improvement. Table 6 effectively isolates the impact of the hyperbolic L2 loss, validating its importance.
However, the authors should clearly differentiate the contribution of the backbone networks from the main innovations of the paper (spherical-based proxy and correspondence-based loss), by comparing with existing methods under the same configurations.

**Weaknesses:**

1. Justification for Spherical Proxy:
Why is the 2-sphere an optimal proxy shape for this task? While the paper adopts the spherical representation, a more thorough explanation of the choice of spherical shapes would be helpful.

2. Size Estimation Despite Normalization: Section 3.1 mentions that the point cloud utilizes normalized coordinates, yet the method predicts the size parameter “s”.  How is accurate size estimation achieved despite normalization? Clarification on this aspect is necessary.

3. Focus on SO(3) Over SE(3): The method predicts rotation “R” and translation “t”, focusing on SO(3) invariance. Why is roto-translation-invariance (SE(3)) not considered? (and scale-invariance) This oversight could limit the method’s robustness in varying spatial contexts.

4. SO(3)-Invariant Features and Rotation Discrimination:
In Section 3.1, if point-wise features are SO(3)-invariant, how does the method differentiate between different rotations? Wouldn't rotation-“invariant” features cause the model to lose rotational information? An explanation on how SO(3)-equivariant features are maintained would be insightful.

5. Shape-Independence and HEALPix Projection:
In section 3.1. HEALPix Spherical projection, does that anchor-based projection guarantee to preserve shape-independence? Even if the spherical projection is already proposed in VI-Net (Lin et al., 2023), ensuring its effectiveness in preserving shape-independence is critical in this framework.

6. Correspondence-Based Loss in Linear Space:
In lines 357-358, the paper claims that learning correspondence-based loss is easier in linear space. Why is this the easier case? Would direct regression provide a better supervisory signal? Providing supporting evidence would strengthen this claim.

7. HEALPix vs. Other Grids: It is convinced that the choice of the HEALPix grid over random rectangular SO(3) grids is explained by the drawbacks of equirectangular grids near the poles.
However, could a random SO(3) grid serve as a viable alternative?
Additionally, in Table 7, evaluating other SO(3) spherical grids, such as SuperFibonacci spirals [A], would provide a comprehensive comparison, given its fast construction properties.

[A] Super-Fibonacci Spirals: Fast, Low-Discrepancy Sampling of SO(3) (Marc Alexa, CVPR 2022)

8. Clarification of this paper’s contribution: Please organize Sections 3 and 4 to focus on the main contribution of this paper.

**Questions:**

1. Rotational Invariance/Equivariance in HEALPix Projection:
How does the HEALPix spherical projection in Section 3.1 maintain rotational invariance or equivariance between input and output?

2. Related Work and Citations: There are two concurrent works leveraging spherical grids for pose estimation—correspondence-based [B] and regression-based [C]. Please cite and discuss these works to position your method within the broader context.

[B] Improving Semantic Correspondence with Viewpoint-Guided Spherical Maps (Mariotti et al., CVPR 2024 ): https://arxiv.org/abs/2312.13216

[C] 3D Equivariant Pose Regression via Direct Wigner-D Harmonics Prediction (Lee et al., NeurIPS 2024 ): https://arxiv.org/abs/2411.00543

---

> ### Author Response · Authors · 2024-11-24
> **Response to Reviewer rB6i (Part 1)**
>
> We appreciate the reviewer for the constructive comments of our work. We hope our following responses can address your concerns.
>
> ------
>
> **S1. Validation of Correspondence Errors on HouseCat6D**
>
> > Good analysis of correspondence errors: The method effectively identifies correspondences, as shown in Table 3, where accurate correspondence significantly enhances pose estimation accuracy. However, consistency of correspondence errors should be validated on another dataset, such as HouseCat6D.
>
> * **Results on HouseCat6D are consistent with those on REAL275.** Thanks for the suggestion. We provide additional comparisons on the HouseCat6D dataset, evaluating the correspondence errors and pose estimation accuracy of our method against the state-of-the-art correspondence-based method, AG-Pose [9]. The results are consistent with those on the REAL275 dataset, demonstrating that the precise correspondence predictions of our approach contribute to superior pose estimation performance.
>
> | Method               | NOCS Angle Error ($^{\circ}$) | 5°2cm | 5°5cm | 10°2cm | 10°5cm |
> | -------------------- | ----------------------------- | ----- | ----- | ------ | ------ |
> | AG-Pose (reproduced) | 40.25                         | 11.7  | 12.9  | 32.7   | 37.2   |
> | SpherePose           | 25.28                         | 19.3  | 25.9  | 40.9   | 55.3   |
>
> ------
>
> **S2. Clarification of Backbone vs. Main Innovations**
>
> > Comprehensive Ablation Studies: The authors provide a detailed analysis of different feature extractors (DINOv2, ColorPointNet++) and loss functions (L2 vs. hyperbolic L2). While not the primary contribution, Table 4 highlights the role of backbone networks in performance improvement. Table 6 effectively isolates the impact of the hyperbolic L2 loss, validating its importance. However, the authors should clearly differentiate the contribution of the backbone networks from the main innovations of the paper (spherical-based proxy and correspondence-based loss), by comparing with existing methods under the same configurations.
>
> * **SpherePose demonstrates superior performance with the same visual backbone.** Thanks for the suggestion. In the revised manuscript, we have updated **Table 4** in the main text to isolate the contribution of backbone networks, highlighting the importance of SO(3)-invariant features. Additionally, we compare our method with existing state-of-the-art approaches under the same visual backbone configuration. SpherePose achieves higher pose estimation accuracy with fewer parameters.
>
> | Method     | Backbone                 | 5°2cm | 5°5cm | 10°2cm | 10°5cm | Parameters (M) | Speed (FPS) |
> | ---------- | ------------------------ | ----- | ----- | ------ | ------ | -------------- | ----------- |
> | SecondPose | DINOv2 & HP-PPF          | 56.2  | 63.6  | 74.7   | 86.0   | 60.2           | 14.3        |
> | AG-Pose    | DINOv2 & PointNet++      | 57.0  | 64.6  | 75.1   | 84.7   | 33.1           | 25.5        |
> | SpherePose | DINOv2 & ColorPointNet++ | 58.2  | 67.4  | 76.2   | 88.2   | 26.5           | 25.3        |
>
> ------
>
> **W1. Justification for Spherical Proxy**
>
> > Justification for Spherical Proxy: Why is the 2-sphere an optimal proxy shape for this task? While the paper adopts the spherical representation, a more thorough explanation of the choice of spherical shapes would be helpful.
>
> * **The spherical proxy is a widely adopted representation.** Due to its intuitive nature in representing object shapes from various viewpoints, many prior works, such as DualPoseNet [10], VI-Net [2], and SecondPose [11], have also utilized spherical representations.
>
> * **The specific choice of proxy shape is not the focus of this work.** Instead, the emphasis lies on leveraging a shared proxy shape to learn shape-independent transformation. Other reasonable proxy shapes may also be applicable, which we leave for future exploration.
>
> ------
>
> **W2. Size Estimation Despite Normalization**
>
> > Size Estimation Despite Normalization:  Section 3.1 mentions that the point cloud utilizes normalized coordinates, yet the method predicts the size parameter “s”. How is accurate size estimation achieved despite normalization? Clarification on this aspect is necessary.
>
> * **Size and translation are pre-estimated.** As described in Section 3.3 ("Inference"), following VI-Net [2], we first utilize a lightweight PointNet++ to predict the translation and size from the raw point cloud. The predicted translation and size are then used to normalize the point cloud, which is subsequently fed into our network for rotation estimation.

---

> ### Author Response · Authors · 2024-11-24
> **Response to Reviewer rB6i (Part 2)**
>
> **W3. Focus on SO(3) Over SE(3)**
>
> > Focus on SO(3) Over SE(3):  The method predicts rotation “R” and translation “t”, focusing on SO(3) invariance. Why is roto-translation-invariance (SE(3)) not considered? (and scale-invariance) This oversight could limit the method’s robustness in varying spatial contexts.
>
> Please refer to our common response **R2** above for more details.
>
> * **Translation estimation is relatively straightforward.** For instance, translation can be initialized as the centroid of the observed point cloud, with a residual predicted subsequently.
> * **Rotation estimation is robust to perturbations in translation.**
> * **Rotation estimation is robust to perturbations in scale.**
>
> ------
>
> **W4. SO(3)-Invariant Features and Rotation Discrimination**
>
> > SO(3)-Invariant Features and Rotation Discrimination: In Section 3.1, if point-wise features are SO(3)-invariant, how does the method differentiate between different rotations? Wouldn't rotation-“invariant” features cause the model to lose rotational information? An explanation on how SO(3)-equivariant features are maintained would be insightful.
>
> Please refer to our common response **R1** above for more details.
>
> * **Rotation estimation network must maintain SO(3)-equivariance between its input and output.**
> * **Correspondence predictor in correspondence-based methods must ensure SO(3)-invariance.**
> * **Features extracted in correspondence-based methods must ensure SO(3)-invariance.**
>
> ------
>
> **W5. Shape-Independence and HEALPix Projection**
>
> > Shape-Independence and HEALPix Projection: In section 3.1. HEALPix Spherical projection, does that anchor-based projection guarantee to preserve shape-independence? Even if the spherical projection is already proposed in VI-Net (Lin et al., 2023), ensuring its effectiveness in preserving shape-independence is critical in this framework.
>
> * **The spherical proxy shape is consistent and shared across different objects, enabling the network to focus solely on shape-independent rotation transformation.** Regardless of the observed object shape, it is projected onto a unit sphere, represented by spherical anchors. This replaces the original object shape with a coherent spherical proxy shape shared across objects, allowing the network to concentrate on learning rotation transformation without being affected by variations in object shapes.
>
> ------
>
> **W6. Correspondence-Based Loss in Linear Space**
>
> > Correspondence-Based Loss in Linear Space: In lines 357-358, the paper claims that learning correspondence-based loss is easier in linear space. Why is this the easier case? Would direct regression provide a better supervisory signal? Providing supporting evidence would strengthen this claim.
>
> * **The ablation studies on different rotation representations in Table 4 of SAR-Net [12] demonstrates that learning is easier in Euclidean space compared to SO(3) space.**
> * **Removing the correspondence loss and directly regressing rotation from features leads to network divergence.** Since the extracted features of our network are SO(3)-invariant, directly regressing rotation from these features suffers from their lack of awareness to rotational information. Even when concatenating SO(3)-equivariant observed coordinates to the point-wise features, the network still fails to converge.
> * **Retaining the correspondence loss and regressing rotation from correspondences leads to reduced performance.** Using a deep estimator, as proposed by DPDN [8], to directly regress rotation from correspondence pairs provides a direct supervisory signal. However, due to its reliance on the precision of correspondence predictions, this approach slightly underperforms compared to fitting rotations using the Umeyama algorithm.
>
> | Rotation Fitting | 5°2cm | 5°5cm | 10°2cm | 10°5cm |
> | ---------------- | ----- | ----- | ------ | ------ |
> | Umeyama          | 58.2  | 67.4  | 76.2   | 88.2   |
> | Deep Estimator   | 57.7  | 66.8  | 76.5   | 88.2   |

---

> ### Author Response · Authors · 2024-11-24
> **Response to Reviewer rB6i (Part 3)**
>
> **W7. HEALPix vs. Other Grids**
>
> > HEALPix vs. Other Grids: It is convinced that the choice of the HEALPix grid over random rectangular SO(3) grids is explained by the drawbacks of equirectangular grids near the poles. However, could a random SO(3) grid serve as a viable alternative? Additionally, in Table 7, evaluating other SO(3) spherical grids, such as SuperFibonacci spirals [A], would provide a comprehensive comparison, given its fast construction properties.
> >
> > [A] Super-Fibonacci Spirals: Fast, Low-Discrepancy Sampling of SO(3) (Marc Alexa, CVPR 2022)
>
> Thanks for the suggestion. Please refer to **Table 7** of our revised manuscript for more details.
>
> * **Random SO(3) grids are not suitable alternatives.** These grids fail to ensure uniform sampling and lack a fixed set of anchor positions, leading to increased storage costs for positional embedding.
> * **We have included comparison with Super-Fibonacci Spirals grids in Table 7.** While Super-Fibonacci Spirals [5] grids outperform Equirectangular grids, the performance is slightly inferior to HEALPix grids.
>
> | Spherical Grids         | 5°2cm | 5°5cm | 10°2cm | 10°5cm |
> | ----------------------- | ----- | ----- | ------ | ------ |
> | HEALPix                 | 58.2  | 67.4  | 76.2   | 88.2   |
> | Equirectangular         | 55.7  | 64.7  | 72.1   | 83.6   |
> | Super-Fibonacci Spirals | 56.5  | 65.0  | 75.8   | 87.2   |
>
> ------
>
> **W8. Clarification of this paper’s contribution**
>
> > Clarification of this paper’s contribution: Please organize Sections 3 and 4 to focus on the main contribution of this paper.
>
> Thanks for the suggestion. We have reorganized **Table 4** in the main text to emphasize the importance of SO(3)-invariant point-wise features rather than backbone networks.
>
> ------
>
> **Q1. Rotational Invariance/Equivariance in HEALPix Projection**
>
> > Rotational Invariance/Equivariance in HEALPix Projection: How does the HEALPix spherical projection in Section 3.1 maintain rotational invariance or equivariance between input and output?
>
> Please refer to our common response **R1** and **R4** above for more details.
>
> * **Rotation estimation network must maintain SO(3)-equivariance between its input and output.**
> * **HEALPix spherical projection maintains SO(3)-equivariance between its input and output.** Specifically, the observed point cloud coordinates in the camera space are projected onto the sphere, resulting in spherical anchor coordinates that remain within the camera space but represent a different shape. This projection preserves SO(3)-equivariance between the observed point cloud and spherical anchors.
> * **Correspondence predictor in correspondence-based methods must ensure SO(3)-invariance.** Subsequently, correspondence-based methods learn the transformation from observed coordinates in the camera space to NOCS coordinates in the canonical object space. This transformation should be SO(3)-invariant.

---

> ### Author Response · Authors · 2024-11-24
> **Response to Reviewer rB6i (Part 4)**
>
> **Q2. Related Work and Citations**
>
> > Related Work and Citations: There are two concurrent works leveraging spherical grids for pose estimation—correspondence-based [B] and regression-based [C]. Please cite and discuss these works to position your method within the broader context.
> >
> > [B] Improving Semantic Correspondence with Viewpoint-Guided Spherical Maps (Mariotti et al., CVPR 2024 ): https://arxiv.org/abs/2312.13216
> >
> > [C] 3D Equivariant Pose Regression via Direct Wigner-D Harmonics Prediction (Lee et al., NeurIPS 2024 ): https://arxiv.org/abs/2411.00543
>
> Thanks for the suggestion. We have discussed the distinctions between our method and them [13, 14] in the **Appendix A.4**. Please refer to our revised manuscript for more details.
>
> ------
>
> **References**
>
> [2] Lin J, Wei Z, Zhang Y, et al. Vi-net: Boosting category-level 6d object pose estimation via learning decoupled rotations on the spherical representations[C]//Proceedings of the IEEE/CVF International Conference on Computer Vision. 2023: 14001-14011.
>
> [6] Alexa M. Super-fibonacci spirals: Fast, low-discrepancy sampling of so (3)[C]//Proceedings of the IEEE/CVF Conference on Computer Vision and Pattern Recognition. 2022: 8291-8300.
>
> [8] Lin J, Wei Z, Ding C, et al. Category-level 6d object pose and size estimation using self-supervised deep prior deformation networks[C]//European Conference on Computer Vision. Cham: Springer Nature Switzerland, 2022: 19-34.
>
> [9] Lin X, Yang W, Gao Y, et al. Instance-adaptive and geometric-aware keypoint learning for category-level 6d object pose estimation[C]//Proceedings of the IEEE/CVF Conference on Computer Vision and Pattern Recognition. 2024: 21040-21049.
>
> [10] Lin J, Wei Z, Li Z, et al. Dualposenet: Category-level 6d object pose and size estimation using dual pose network with refined learning of pose consistency[C]//Proceedings of the IEEE/CVF International Conference on Computer Vision. 2021: 3560-3569.
>
> [11] Chen Y, Di Y, Zhai G, et al. Secondpose: Se (3)-consistent dual-stream feature fusion for category-level pose estimation[C]//Proceedings of the IEEE/CVF Conference on Computer Vision and Pattern Recognition. 2024: 9959-9969.
>
> [12] Lin H, Liu Z, Cheang C, et al. Sar-net: Shape alignment and recovery network for category-level 6d object pose and size estimation[C]//Proceedings of the IEEE/CVF conference on computer vision and pattern recognition. 2022: 6707-6717.
>
> [13] Mariotti O, Mac Aodha O, Bilen H. Improving semantic correspondence with viewpoint-guided spherical maps[C]//Proceedings of the IEEE/CVF Conference on Computer Vision and Pattern Recognition. 2024: 19521-19530.
>
> [14] Lee J, Cho M. 3D Equivariant Pose Regression via Direct Wigner-D Harmonics Prediction[C]//The Thirty-eighth Annual Conference on Neural Information Processing Systems.

---

> ### Author Response · Authors · 2024-11-27
> **Looking Forward to Feedback**
>
> We sincerely thank you for your time in reviewing our paper and your constructive comments. We have posted our point-to-point responses in the review system. Since the public discussion phase will end soon, we appreciate if you could read our responses and let us know your feedback. Please let us know if you have any further questions or if our clarifications and paper updates are satisfactory for an improved rating.

---

> ### Comment · Reviewer_rB6i · 2024-11-27
>
> The author rebuttal has successfully addressed my concerns regarding S1, S2, W3, W4, W6, and W7. Additionally, please ensure that the manuscript updates revisions based on Q1 and Q2.
>
> However, the justification for using a spherical proxy in this paper is not sufficiently established solely based on its usage in other works (W1). The reliance on off-the-shelf models for estimating $S$ increases the overall model complexity (W2). Furthermore, when projecting onto the HEALPix-based spherical proxy, there is no clear guarantee that objects of the same category will be consistently projected (W5).
>
> Despite the unclearness in W1, W2, and W5, my concerns regarding S1, S2, W3, W4, W6, W7, Q1, and Q2 have been resolved. Therefore, I am raising my score.

---

> ### Author Response · Authors · 2024-11-27
> **Response to Further Concerns by Reviewer rB6i**
>
> We sincerely appreciate the reviewer's support for the acceptance of our work. As suggested, we have incorporated additional explanations and discussions regarding **Q1** (in the **Appendix A.1**) and **Q2** (in the **Appendix A.4**) in the revised manuscript. For the remaining concerns, **W1**, **W2**, and **W5**, we have further posted our point-to-point responses. We hope our replies effectively resolve your concerns. If you have any further questions or suggestions, we would be grateful for your comments to help us improve our work.
>
> ------
>
> **W1. Justification for Spherical Proxy**
>
> > Justification for Spherical Proxy: Why is the 2-sphere an optimal proxy shape for this task? While the paper adopts the spherical representation, a more thorough explanation of the choice of spherical shapes would be helpful.
>
> > The justification for using a spherical proxy in this paper is not sufficiently established solely based on its usage in other works.
>
> The core insight of this work is to learn shape-independent transformation using a shared proxy shape. To achieve this, it is crucial to select an appropriate proxy shape that satisfies two key properties:
>
> 1. **Uniform Sampling**: The proxy shape should capture the object shapes as uniformly as possible.
> 2. **Consistency of Sampling Density Under Rotation**: The proxy shape must ensure that the sampling density remains consistent for a certain object part under different object rotations. This is important to avoid introducing a biased distribution of sampled data when the object is rotated.
>
> For a 2-sphere, the use of uniformly sampled grids, such as the HEALPix grids, achieves area-uniform sampling. Furthermore, due to the isotropic nature of a sphere, the sampling density remains consistent under varying object rotations. For instance, the grids responsible for capturing the camera lens structure will remain within a fixed spherical cone, and its corresponding sampling density on the sphere will not change, regardless of the camera’s rotation.
>
> In contrast, for other classic proxy shapes like ellipsoids, cylinders, or cubes, although it is possible to achieve uniform sampling through specific rules (e.g., equidistant grids on the surface of a cube), these shapes are not isotropic. The sampling density within a particular viewing cone for a certain object part will vary under different object rotations. For instance, when the camera lens rotates from facing a flat face of a cube to facing one of its corners, the corresponding sampling density on the cube will change. This can lead to biased distribution of the sampled data, making them less suitable for capturing the point cloud structures.
>
> ------
>
> **W2. Size Estimation Despite Normalization**
>
> > Size Estimation Despite Normalization:  Section 3.1 mentions that the point cloud utilizes normalized coordinates, yet the method predicts the size parameter “s”. How is accurate size estimation achieved despite normalization? Clarification on this aspect is necessary.
>
> > The reliance on off-the-shelf models for estimating S increases the overall model complexity.
>
> The translation and size are estimated using a lightweight PointNet++ network, introducing minimal additional model complexity.
>
> | Network                        | Parameters (M) |
> | ------------------------------ | -------------- |
> | Rotation Estimator             | 24.9           |
> | Translation and Size Estimator | 1.6            |
> | SpherePose                     | 26.5           |
>
> ------
>
> **W5. Shape-Independence and HEALPix Projection**
>
> > Shape-Independence and HEALPix Projection: In section 3.1. HEALPix Spherical projection, does that anchor-based projection guarantee to preserve shape-independence? Even if the spherical projection is already proposed in VI-Net (Lin et al., 2023), ensuring its effectiveness in preserving shape-independence is critical in this framework.
>
> > When projecting onto the HEALPix-based spherical proxy, there is no clear guarantee that objects of the same category will be consistently projected.
>
> Indeed, objects of the same category with different shapes cannot guarantee consistent projections. For example, when cameras with varying lens lengths project onto a sphere, the cone of the lens corresponding to the sphere will differ. However, the primary goal of our spherical proxy shape design is to decouple shape variations from the transformation learning process. Since the transformation exists between the unit spheres in camera and object spaces, the network does not need to focus on the specific object shapes. Instead, it only needs to focus on the rotation information. Therefore, even though the size of the cone corresponding to the camera lens may vary, its direction remains consistent and capable of encoding rotation information.

---

### Official Review · Reviewer_LyLo · 2024-11-04

**Soundness:** 3
**Presentation:** 3
**Contribution:** 3
**Rating:** 8
**Confidence:** 4

**Summary:**

The paper addresses the challenge of object pose estimation in 3D space, specifically overcoming the limitations of existing methods that rely heavily on 3D model shapes. It proposes an approach called SpherePose, which utilizes spherical representations to create shape-independent transformations, thereby improving correspondence prediction accuracy. The method incorporates three core innovations: SO(3)-invariant feature extraction, spherical feature interaction using attention mechanisms, and a hyperbolic correspondence loss function for precise supervision. This paper mainly introduces a new proxy shape for objects and a robust architecture for category-level pose estimation. Empirical validation shows superior performance against state-of-the-art methods.

**Strengths:**

- This paper uses a sphere as a proxy to implement category-level object pose estimation. It transforms the 3D shape to a uniform sphere via HEALPix spherical representations, which leads the network to focus on the semantic consistency between different objects instead of shape deviation.
- The architecture for category-level object pose estimation that achieves precise different objects in one category correspondence prediction.
- The experiments show that the proposed methods achieve the SOTA results on 6D pose estimation tasks in several datasets.

**Weaknesses:**

- The point cloud patches are occluded. The center of a point cloud patch may not be the corresponding object's center. The projection results changed a lot when the center moved. Besides, different object point cloud patches have various occlusions. The correspondence by the spherical proxy, whether a real semantic correspondence of objects, could have more evidence or qualitative results.
- The positional encoding of the anchors needs more explanation. The way to describe the anchor position is vague, spherical coordinates or something else. The position of an anchor could influence the rotation invariance, making the results sensitive to rotation.

**Questions:**

- Given that the method relies on SO(3)-invariant feature extraction and RGB-based features, how robust is the approach to diverse lighting conditions and texture variations across different objects?
- Can the spherical feature interaction using attention mechanisms generalize well to objects with high self-occlusion or cluttered environments, and how is this tested?
- The hyperbolic correspondence loss function is designed to improve gradient behavior near zero. How does this compare quantitatively with traditional loss functions in terms of convergence speed and accuracy?

---

> ### Author Response · Authors · 2024-11-24
> **Response to Reviewer LyLo**
>
> We appreciate the reviewer for the constructive comments of our work. We hope our following responses can address your concerns.
>
> ------
>
> **W1. Point Cloud Occlusion and Center Alignment**
>
> > The point cloud patches are occluded. The center of a point cloud patch may not be the corresponding object's center. The projection results changed a lot when the center moved. Besides, different object point cloud patches have various occlusions. The correspondence by the spherical proxy, whether a real semantic correspondence of objects, could have more evidence or qualitative results.
>
> Please refer to our common response **R2** and **R3** above for more details.
>
> * **Translation estimation is relatively straightforward.** For instance, translation can be initialized as the centroid of the observed point cloud, with a residual predicted subsequently.
> * **Rotation estimation is robust to perturbations in translation.**
> * **Spherical feature interaction demonstrates a certain degree of generalization to varying levels of occlusion.**
>
> ------
>
> **W2. Explanation about Anchor Position Encoding**
>
> > The positional encoding of the anchors needs more explanation. The way to describe the anchor position is vague, spherical coordinates or something else. The position of an anchor could influence the rotation invariance, making the results sensitive to rotation.
>
> Please refer to our common response **R4** above for more details.
>
> * **Spherical anchor positions are defined as the unit spherical coordinates of grid centers.**
>
> * **Spherical anchor position encoding does not affect rotation invariance, as it is not injected into the anchor features.**
>
> -------
>
> **Q1. Robustness to Visual Variations**
>
> > Given that the method relies on SO(3)-invariant feature extraction and RGB-based features, how robust is the approach to diverse lighting conditions and texture variations across different objects?
>
> * **During training, we apply data augmentation on RGB colors.** Specifically, following prior works such as DPDN [8] and VI-Net [2], we enhance the robustness to diverse visual variations by applying color perturbations to RGB images using PyTorch: $\operatorname{ColorJitter}(\textrm{brightness} = 0.2, \textrm{contrast} = 0.2, \textrm{saturation} = 0.2, \textrm{hue} = 0.05)$.
>
> - **During testing, our method demonstrates robustness to color perturbations.** The table below provides results under various levels of color perturbations applied to RGB images during testing. This robustness can also be partially attributed to the generalization capability of the large-scale pretrained vision model, DINOv2 [7].
>
> | Color Perturbation During Testing | 5°2cm | 5°5cm | 10°2cm | 10°5cm |
> | --------------------------------- | ----- | ----- | ------ | ------ |
> | None                              | 58.2  | 67.4  | 76.2   | 88.2   |
> | (0.2, 0.2, 0.2, 0.05)             | 57.9  | 67.1  | 76.2   | 88.0   |
> | (0.5, 0.5, 0.5, 0.2)              | 56.0  | 65.8  | 74.8   | 87.4   |
>
> ------
>
> **Q2. Generalization to Occluded Objects**
>
> > Can the spherical feature interaction using attention mechanisms generalize well to objects with high self-occlusion or cluttered environments, and how is this tested?
>
> Please refer to our common response **R3** above for more details.
>
> * **Spherical feature interaction demonstrates a certain degree of generalization to varying levels of occlusion.** However, when the occlusion ratio is too high, spherical feature interaction struggles to reason about the features of heavily occluded anchors due to the extremely limited available object information.
>
> ------
>
> **Q3. Hyperbolic Correspondence Loss Comparison**
>
> > The hyperbolic correspondence loss function is designed to improve gradient behavior near zero. How does this compare quantitatively with traditional loss functions in terms of convergence speed and accuracy?
>
> Please refer to **Figure 7** in the **Appendix** of our revised manuscript for more details.
>
> * .**The hyperbolic correspondence loss function not only accelerates convergence but also achieves higher accuracy.**
>
> ------
>
> **References**
>
> [2] Lin J, Wei Z, Zhang Y, et al. Vi-net: Boosting category-level 6d object pose estimation via learning decoupled rotations on the spherical representations[C]//Proceedings of the IEEE/CVF International Conference on Computer Vision. 2023: 14001-14011.
>
> [7] Oquab M, Darcet T, Moutakanni T, et al. DINOv2: Learning Robust Visual Features without Supervision[J]. Transactions on Machine Learning Research.
>
> [8] Lin J, Wei Z, Ding C, et al. Category-level 6d object pose and size estimation using self-supervised deep prior deformation networks[C]//European Conference on Computer Vision. Cham: Springer Nature Switzerland, 2022: 19-34.

---

> ### Author Response · Authors · 2024-11-27
> **Looking Forward to Feedback**
>
> We sincerely thank you for your time in reviewing our paper and your constructive comments. We have posted our point-to-point responses in the review system. Since the public discussion phase will end soon, we appreciate if you could read our responses and let us know your feedback and further comments.

---

### Author Response · Authors · 2024-11-24
**Common Response to All Reviewers**

We thank all reviewers for their thoughtful comments and valuable feedback. Here, we provide a common response to some of the recurring issues, and summarize the revisions made to the manuscript according to the reviewers' suggestions, along with the references cited during the response process. Reviewer-specific responses are left as direct comments.

------

**Summary of Changes**

* As suggested by **Reviewer p4nq**, we have refined the description about ColorPointNet++ and DINOv2 features in **Section 1** and **Section 3**.
* As suggested by **Reviewer rB6i**, we have reorganized **Table 4** in Section 4 to emphasize the importance of SO(3)-invariant point-wise features.
* As suggested by **Reviewer rB6i**, we have included comparison with Super-Fibonacci Spirals [6] grids in **Table 7** in Section 4.
* As suggested by **Reviewer rB6i** and **Reviewer p4nq**, we have included a detailed explanation of SO(3)-invariance/equivariance in the **Appendix A.1**.
* As suggested by **Reviewer p4nq**, we have included an analysis of the SO(3)-invariance of DINOv2 [7] features in **Figure 5** in the Appendix.
* As suggested by **Reviewer v4xt**, we have included results that exclude RGB or radius values in **Table 10** in the Appendix.
* As suggested by **Reviewer LyLo**, we have included comparison of convergence speed and accuracy for the hyperbolic correspondence loss function in **Figure 7** in the Appendix.
* As suggested by **Reviewer rB6i**, we have discussed the distinctions between our method and two concurrent works [13, 14] in the **Appendix A.4**.
* We have made minor adjustments to the structure of the Appendix for better readability.
* We have refined the alignment of tables to enhance clarity and visual appeal.

------

**References**

Here are references cited in all responses:

[1] Umeyama S. Least-squares estimation of transformation parameters between two point patterns[J]. IEEE Transactions on Pattern Analysis & Machine Intelligence, 1991, 13(04): 376-380.

[2] Lin J, Wei Z, Zhang Y, et al. Vi-net: Boosting category-level 6d object pose estimation via learning decoupled rotations on the spherical representations[C]//Proceedings of the IEEE/CVF International Conference on Computer Vision. 2023: 14001-14011.

[3] Qi C R, Yi L, Su H, et al. Pointnet++: Deep hierarchical feature learning on point sets in a metric space[J]. Advances in neural information processing systems, 2017, 30.

[4] Gorski K M, Hivon E, Banday A J, et al. HEALPix: A framework for high-resolution discretization and fast analysis of data distributed on the sphere[J]. The Astrophysical Journal, 2005, 622(2): 759.

[5] Wang H, Sridhar S, Huang J, et al. Normalized object coordinate space for category-level 6d object pose and size estimation[C]//Proceedings of the IEEE/CVF Conference on Computer Vision and Pattern Recognition. 2019: 2642-2651.

[6] Alexa M. Super-fibonacci spirals: Fast, low-discrepancy sampling of so (3)[C]//Proceedings of the IEEE/CVF Conference on Computer Vision and Pattern Recognition. 2022: 8291-8300.

[7] Oquab M, Darcet T, Moutakanni T, et al. DINOv2: Learning Robust Visual Features without Supervision[J]. Transactions on Machine Learning Research.

[8] Lin J, Wei Z, Ding C, et al. Category-level 6d object pose and size estimation using self-supervised deep prior deformation networks[C]//European Conference on Computer Vision. Cham: Springer Nature Switzerland, 2022: 19-34.

[9] Lin X, Yang W, Gao Y, et al. Instance-adaptive and geometric-aware keypoint learning for category-level 6d object pose estimation[C]//Proceedings of the IEEE/CVF Conference on Computer Vision and Pattern Recognition. 2024: 21040-21049.

[10] Lin J, Wei Z, Li Z, et al. Dualposenet: Category-level 6d object pose and size estimation using dual pose network with refined learning of pose consistency[C]//Proceedings of the IEEE/CVF International Conference on Computer Vision. 2021: 3560-3569.

[11] Chen Y, Di Y, Zhai G, et al. Secondpose: Se (3)-consistent dual-stream feature fusion for category-level pose estimation[C]//Proceedings of the IEEE/CVF Conference on Computer Vision and Pattern Recognition. 2024: 9959-9969.

[12] Lin H, Liu Z, Cheang C, et al. Sar-net: Shape alignment and recovery network for category-level 6d object pose and size estimation[C]//Proceedings of the IEEE/CVF conference on computer vision and pattern recognition. 2022: 6707-6717.

[13] Mariotti O, Mac Aodha O, Bilen H. Improving semantic correspondence with viewpoint-guided spherical maps[C]//Proceedings of the IEEE/CVF Conference on Computer Vision and Pattern Recognition. 2024: 19521-19530.

[14] Lee J, Cho M. 3D Equivariant Pose Regression via Direct Wigner-D Harmonics Prediction[C]//The Thirty-eighth Annual Conference on Neural Information Processing Systems.

---

> ### Author Response · Authors · 2024-11-24
> **Common Response to All Reviewers (Part 1)**
>
> **R1. Clarifications on Rotation-Invariance/Equivariance**
>
> * **Rotation estimation network must maintain SO(3)-equivariance between its input and output.** The rotation estimation task is formulated as $R = \operatorname{Net}(P)$, where the input $P \in \mathbb{R}^{N \times 3}$ is the observed point cloud, and the output $R \in \mathrm{SO(3)}$ is the estimated rotation. The SO(3)-equivariance between input and output is formulated as: $\operatorname{Net}(\psi_G(P)) = \psi_G( \operatorname{Net}(P) )$.
>
> * **Correspondence predictor in correspondence-based methods must ensure SO(3)-invariance.** This group of methods can be expressed as $R = \operatorname{Net}(P) = \Phi(P, \operatorname{Corr}(P))$, where $\Phi$ is the rotation solver and $\operatorname{Corr}$ is the correspondence (NOCS) predictor. The SO(3)-equivariance between the network input and output is formulated as: $\Phi(\psi_G(P), \operatorname{Corr}(\psi_G(P))) = \psi_G( \Phi(P, \operatorname{Corr}(P)) )$.
>   * The primary target of these methods is to predict the NOCS coordinates for each observed point and then solve the final rotation using the point-wise correspondences between camera coordinates $P$ and canonical object coordinates $\operatorname{Corr}(P) \in \mathbb{R}^{N \times 3}$ via the rotation solver (e.g., the Umeyama [1] algorithm).
>   * Given fixed reference coordinates $\operatorname{Corr}(P)$, the rotation solver is SO(3)-equivariant with respect to the other input $P$: $\Phi (\psi_G(P), \operatorname{Corr}(P)) = \psi_G(\Phi (P, \operatorname{Corr}(P)))$. Consequently, the correspondence predictor needs to be designed to ensure SO(3)-invariance: $\operatorname{Corr}(\psi_G(P)) = \operatorname{Corr}(P)$. It means that no matter how an object is rotated, a specific point on that object should be mapped to a static coordinate in NOCS.
> * **Features extracted in correspondence-based methods must ensure SO(3)-invariance.** Since the NOCS predictor is defined as $\operatorname{Corr}(P) = \operatorname{MLP}(f(P))$, where $f(P)$ is the point-wise feature extractor, $f(P)$ also needs to be SO(3)-invariant.
>
> ------
>
> **R2. Robustness to Translation and Scale**
>
> * **Translation estimation is relatively straightforward.** For instance, translation can be initialized as the centroid of the observed point cloud, with a residual predicted subsequently. Following VI-Net [2], we utilize a lightweight PointNet++ [3] for translation prediction, which already achieves sufficient accuracy.
>
> | Translation Error | < 2cm | < 5cm |
> | ----------------- | ----- | ----- |
> | SpherePose        | 80.5% | 97.2% |
>
> * **Rotation estimation is robust to perturbations in translation.** The rotation estimation network first centralizes the observed point cloud using the predicted translation. When the translation used for centralizing is perturbed, the rotation estimation performance remains mostly unaffected.
>
> | Translation Perturbation | 5°2cm | 5°5cm | 10°2cm | 10°5cm |
> | ------------------------ | ----- | ----- | ------ | ------ |
> | None                     | 58.2  | 67.4  | 76.2   | 88.2   |
> | < 2cm                    | 57.9  | 67.1  | 76.1   | 87.9   |
> | < 5cm                    | 55.8  | 64.7  | 75.3   | 87.1   |
>
> * **Rotation estimation is robust to perturbations in scale.** When the scale used for normalization is perturbed, the rotation estimation performance also exhibits minimal impact.
>
> | Scale Perturbation | 5°2cm | 5°5cm | 10°2cm | 10°5cm |
> | ------------------ | ----- | ----- | ------ | ------ |
> | None               | 58.2  | 67.4  | 76.2   | 88.2   |
> | [0.8, 1.25]        | 58.0  | 67.1  | 76.1   | 88.0   |
> | [0.5, 2.0]         | 55.8  | 64.4  | 74.3   | 86.1   |

---

> ### Author Response · Authors · 2024-11-24
> **Common Response to All Reviewers (Part 2)**
>
> **R3. Generalization to Occluded Objects**
>
> * **Spherical feature interaction aids in reasoning about the features of occluded anchors.** As shown in the table, using correspondences from even only occluded anchors achieves fitting accuracy comparable to that of using only visible anchors. This result indicates that spherical feature interaction enables reasoning about the features of occluded anchors from a holistic perspective, leading to consistent correspondence predictions. In our approach, correspondences from all anchors are used by default for rotation fitting.
>
> | Correspondences for Rotation Fitting | 5°2cm | 5°5cm | 10°2cm | 10°5cm |
> | ------------------------------------ | ----- | ----- | ------ | ------ |
> | from All Spherical Anchors           | 58.2  | 67.4  | 76.2   | 88.2   |
> | from Visible Spherical Anchors       | 58.2  | 67.4  | 76.3   | 88.2   |
> | from Occluded Spherical Anchors      | 58.1  | 67.3  | 76.2   | 88.2   |
>
> * **Spherical feature interaction demonstrates a certain degree of generalization to varying levels of occlusion.** First, we analyze the distribution of object visibility ratios on the REAL275 dataset, observing that most objects have more than half of their parts occluded, with visible ratios typically between 0.3 and 0.5. Next, we limit the max visible ratio for all objects, and the results show that our method maintains good generalization performance when the visible ratio exceeds 0.3. However, when the visible ratio drops significantly, such as to 0.1, the limited available object information hampers spherical feature interaction, making it challenging to reason about the features of the heavily occluded anchors.
>
> | Visible Ratio Distribution | [0.0, 0.1] | (0.1, 0.2] | (0.2, 0.3] | (0.3, 0.4] | (0.4, 0.5] | (0.5, 1.0] |
> | -------------------------- | ---------- | ---------- | ---------- | ---------- | ---------- | ---------- |
> | REAL275                    | 0.20%      | 0.26%      | 6.59%      | 40.73%     | 45.12%     | 7.10%      |
>
> | Max Visible Ratio | 5°2cm | 5°5cm | 10°2cm | 10°5cm |
> | ----------------- | ----- | ----- | ------ | ------ |
> | Raw               | 58.2  | 67.4  | 76.2   | 88.2   |
> | 0.5               | 58.1  | 67.3  | 76.2   | 88.2   |
> | 0.4               | 58.0  | 67.1  | 76.4   | 88.3   |
> | 0.3               | 56.4  | 65.6  | 76.4   | 88.4   |
> | 0.2               | 53.1  | 61.5  | 75.3   | 87.3   |
> | 0.1               | 39.4  | 45.0  | 69.3   | 80.7   |
>
> ------
>
> **R4. Details about Spherical Anchors and NOCS Coordinates**
>
> * **Spherical anchor positions are defined as the unit spherical coordinates of grid centers.** Specifically, the unit sphere is first divided into $M$ equal-area grids using the HEALPix [4] grids. The unit sphere coordinate of each grid center is then regarded as the corresponding anchor position $A_m \in \mathbb{R}^3$.
> * **Spherical anchor position encoding does not affect rotation invariance, as it is not injected into the anchor features.** Instead, the role of anchor position encoding is to provide relative positional information during spherical feature interaction to assist in attention computation. Thus, anchor position encoding only influences the *query* and *key* in the attention mechanism, leaving the *value* unchanged, thereby preserving rotation invariance: $Q=F+E^{pos},K=F+E^{pos},V=F$.
> * **Spherical NOCS coordinates are inherently unit vectors.** Since the spherical anchor positions are defined as unit vectors on the unit sphere, applying the ground-truth rotation transformation to these anchors results in corresponding NOCS coordinates that remain unit vectors. Therefore, the predicted NOCS coordinates are also normalized to unit vectors. Regarding the original definition of NOCS [5] within a unit cube, which constrains the coordinates with a range of [−0.5, 0.5], we can easily adapt by scaling the unit sphere to have a radius of 0.5, aligning with this convention.

---

### Meta-Review · Area_Chair_6GB2 · 2024-12-23

**Metareview:**

The paper addresses the task of object pose estimation in 3D space, overcoming the limitations of existing methods that rely heavily on 3D model shapes. The proposed method uses a sphere as a shared proxy shape for objects, enabling the learning of shape-independent transformations from spherical representations. To enhance the precision of correspondences on the sphere, it incorporates three core components, including SO(3)-invariant point-wise feature extraction, spherical feature interaction, and a hyperbolic correspondence loss function. Empirical validation shows superior performance against state-of-the-art methods.
All reviewers appreciated the novelty of the proposed approach and its good empirical performance. The main concerns were unclear writings/details and missing experiments/analyses. The authors’ detailed rebuttal addressed most of them, resulting in unanimous acceptance at the end of the discussion. AC thus recommends acceptance.

**Additional Comments On Reviewer Discussion:**

The main concerns were unclear writings/details and missing experiments/analyses. The authors’ rebuttal addressed most of them, providing detailed point-to-point explanations and additional experiments.

---

### Decision · Program_Chairs · 2025-01-22

Accept (Poster)